# Directed Graph Contrastive Learning

**Zekun Tong**[1]   **Yuxuan Liang**[1]   **Henghui Ding** [2,3,*]
**Yongxing Dai** [4]   **Xinke Li**[1]   **Changhu Wang**[2]
[1]National University of Singapore   [2]ByteDance   [3]ETH Zürich   [4]Peking University
{zekuntong,liangyuxuan,xinke.li}@u.nus.edu
henghui.ding@vision.ee.ethz.ch, yongxingdai@pku.edu.cn
changhu.wang@bytedance.com

## Abstract

Graph Contrastive Learning (GCL) has emerged to learn generalizable representations from contrastive views. However, it is still in its infancy with two concerns: 1) changing the graph structure through data augmentation to generate contrastive views may mislead the message passing scheme, as such graph changing action deprives the intrinsic graph structural information, especially the directional structure in directed graphs; 2) since GCL usually uses predefined contrastive views with hand-picking parameters, it does not take full advantage of the contrastive information provided by data augmentation, resulting in incomplete structure information for models learning. In this paper, we design a directed graph data augmentation method called Laplacian perturbation and theoretically analyze how it provides contrastive information without changing the directed graph structure. Moreover, we present a directed graph contrastive learning framework, which dynamically learns from all possible contrastive views generated by Laplacian perturbation. Then we train it using multi-task curriculum learning to progressively learn from multiple easy-to-difficult contrastive views. We empirically show that our model can retain more structural features of directed graphs than other GCL models because of its ability to provide complete contrastive information. Experiments on various benchmarks reveal our dominance over the state-of-the-art approaches.

## 1 Introduction

There is a growing interest in learning from **directed graphs** using Graph Neural Networks (GNNs) [17, 12, 18, 50] to tackle practical problems, such as time-series problems [1, 4, 49], fine-grained traffic prediction [29, 21] and recommendation system [35, 7, 40]. Current GNNs designed for learning from directed graphs [23, 43, 44] are trained end-to-end under supervision. This training scheme shows excellent performance by virtue of enough labeled data. Correspondingly, to make use of rich unlabeled data, several Graph Contrastive Learning (GCL) [39, 63, 69, 34, 13, 67] works are proposed based on GNNs and Contrastive Learning (CL) [5, 45, 14]. They utilize data augmentation methods to generate contrastive views from the original graphs, then force views generated from the same instance (node or graph) closer while views from different instances apart using *InfoNCE-like* [28] objective function. However, current GCL methods encounter some issues in processing directed graphs, mainly in *data augmentation method* and *contrastive learning framework*.

Firstly, most *data augmentation* methods used in GCL [63, 69, 68] do not take the directed graph structure into account and may discard distinctive direction information. For example, the idea of dropping nodes/edges [68, 63, 52] is borrowed from random erasing used in images [66], which overlooks the discrepancy of nodes and edges in different graph structures. Moreover, by grasping contrastive information through changing graph structure, part of the distinctive direction information, *e.g.,* irreversible time-series relationships, will inevitably be lost. The message passing scheme will

---

*Corresponding author

35th Conference on Neural Information Processing Systems (NeurIPS 2021).

also be mislead, making it difficult for GNN-based encoders to learn from directed graphs. There is a lack of data augmentation methods that are specifically designed for directed graphs to retain the original directed graph structure while providing enough contrastive information.

Besides, common *contrastive learning frameworks* are not optimized for directed graphs, they can only learn from a limited number of contrastive views [63, 69, 13]. However, due to the complex structure of directed graphs, it is insufficient to use a small number of contrastive views to fully understand their structural characteristics [43]. Furthermore, since the contrastive views have to be determined before training, it becomes another problem to select the ideal views for the downstream task during pre-processing, about which we actually have no knowledge in advance [42, 59]. Hand-picking contrastive views also cause a decrease in generalization. Several works [41, 61] try to obtain more information by increasing the number of contrastive views. However, they still have to pre-define the views and trade off between the number of views and the training difficulty.

To address these two issues, we first propose a directed graph data augmentation method called **Laplacian perturbation**. Since the contrastive views are passed to the encoder in the form of Laplacian matrix, a desirable data augmentation method is to perturb the Laplacian matrix without changing the directed graph structure. To achieve this, we adopt the approximate directed graph Laplacian [43] where a teleport probability is introduced to control the degree of approximation. Thus, by adding a perturbation term to this probability, we can simply augment the Laplacian matrix without altering the directed graph structure. As it is time-consuming to calculate the Laplacian matrix, we then speed it up using the power method [24]. In addition, from the theoretical analysis, we find that the Laplacian perturbation is essentially a perturbation to the directed graph entropy. The perturbation term essentially determines the magnitude of the entropy perturbation error, which affects the magnitude of the contrastive information.

To learn the complete structural characteristics of directed graphs, we design a **directed graph contrastive learning framework** to learn from contrastive views generated by Laplacian perturbation. First, we introduce a generalized dynamic-view contrastive objective which maximizes the sum of the mutual information between the representations of all contrastive views. This objective allows the contrastive views to change dynamically during training and motivates the encoder to learn a generalized representation from all views provided by data augmentation. However, this dynamic-view objective function is hard to optimize. Thus, we leverage the multi-task curriculum learning strategy [33, 36, 10, 26] to divide multiple contrastive views into sub-tasks with various difficulties and progressively learn from easy-to-difficult sub-tasks. In this way, we can learn all views step-by-step to reach the final objective. Moreover, learning from all contrastive views eliminates the need to adjust the data augmentation parameters anymore.

We empirically show that our **Di**rected **G**raph **C**ontrastive **L**earning (**DiGCL**) outperforms the competitive baselines in the settings of unsupervised and supervised learning. Systematic analysis is also carried out to analyze the performance of various augmentations on the mainstream benchmarks and the impact of different pacing functions on the performance of directed graph contrastive learning.

## 2 Directed Graph Data Augmentation

In this section, we first design a directed graph data augmentation scheme named Laplacian perturbation. As it is time-consuming to calculate the Laplacian matrix, we then speed it up using the power method [24]. Finally, we analyze the proposed data augmentation scheme theoretically.

### 2.1 Directed Graph Laplacian and its Approximation

Formally, let a directed graph $\mathcal{G} = (\mathcal{V}, \mathcal{E})$, its adjacency matrix can be denoted as $\mathbf{A} = \{0,1\}^{n \times n}$, where $n = |\mathcal{V}|$. The nodes are described by the feature matrix $\mathbf{X} \in \mathbb{R}^{n \times c}$, with the number of features $c$ per node. Intuitively, the data augmentation can be performed at the topological- and feature-level, operating on $\mathbf{A}$ and $\mathbf{X}$, respectively [63, 69]. The distinctive attribute of directed graphs is the directionality of the edges, which leads us to focus on *topological node-level* data augmentation.

The augmented directed graph is fed to the GNN-based encoder in the form of Laplacian matrix to learn the representation. Since $\mathcal{G}$ may contain isolated nodes or could be formed into bipartite graph, it is not appropriate to trivially use directed graph Laplacian [6]. To relax this constraint, we use its

approximate form: the approximate directed graph Laplacian matrix [43], which is defined as

$$\mathbf{L}_{\text{appr}}(\mathcal{G}, \alpha) = \mathbf{I} - \frac{1}{2}\left(\mathbf{\Pi}_{\text{appr}}^{\frac{1}{2}}\tilde{\mathbf{P}}\mathbf{\Pi}_{\text{appr}}^{-\frac{1}{2}} + \mathbf{\Pi}_{\text{appr}}^{-\frac{1}{2}}\tilde{\mathbf{P}}^T\mathbf{\Pi}_{\text{appr}}^{\frac{1}{2}}\right), \tag{1}$$

where $\mathbf{\Pi}_{\text{appr}} = \frac{1}{\|\pi_{\text{appr}}\|_1}\text{Diag}(\pi_{\text{appr}})$. The approximate eigenvector $\pi_{\text{appr}}$ is defined as

$$(1-\alpha)\pi_{\text{appr}}\tilde{\mathbf{P}} + \frac{1}{n}\frac{\alpha}{1+\alpha}\mathbf{1}^{1\times n} = \pi_{\text{appr}}, \tag{2}$$

where $\tilde{\mathbf{P}} = \tilde{\mathbf{D}}^{-1}\tilde{\mathbf{A}}$, $\tilde{\mathbf{A}} = \mathbf{A} + \mathbf{I}^{n\times n}$ denotes the transition matrix with added self-loops and the diagonal degree matrix $\tilde{\mathbf{D}}(u,u) = \sum_{v\in\mathcal{V}}\tilde{\mathbf{A}}(u,v)$. Tong *et al.* [43] add self-loops to the original directed graph to ensure $\mathcal{G}$ to be *aperiodic* and redefine the transition matrix based on *personalized* PageRank with the teleport probability $\alpha \in (0,1)$ to guarantee the redefined matrix to be *irreducible*. Note that we follow the [43] and set $\alpha = 0.1$ in this paper.

According to Eq. (1), existing data augmentation methods [63, 69], *e.g., node/edge dropping, subgraph sampling*, need to obtain the contrastive information by changing $\tilde{\mathbf{P}}$. Their use of sampling-based methods inevitably corrupts the directed graph structure and thus misleads the message passing in the GNN-based encoder. A desirable data augmentation method on $\mathbf{L}_{\text{appr}}$ is to perturb $\pi_{\text{appr}}$ reasonably without altering the directed graph structure $\tilde{\mathbf{P}}$[1].

## 2.2 Directed Draph Data Augmentation with Laplacian Perturbation

From Eq. (2), it is easy to find that the eigenvector $\pi_{\text{appr}}$ depends on the $\tilde{\mathbf{P}}$ and $\alpha$. In other words, we can shift the teleport probability $\alpha$ without changing the directed graph structure $\tilde{\mathbf{P}}$, thus altering the eigenvector $\pi_{\text{appr}}$ and finally perturbing the Laplacian matrix $\mathbf{L}_{\text{appr}}$.

**DEFINITION 1** *Laplacian perturbation. Given a directed graph $\mathcal{G} = (\mathcal{V}, \mathcal{E})$ and the teleport probability $\alpha$, we define the Laplacian perturbation opertation $\Phi(\cdot)$ on the $\mathbf{L}_{\text{appr}}(\tilde{\mathbf{P}}, \alpha)$ as*

$$\Phi_{\Delta\alpha}(\mathcal{G}, \alpha) = \mathbf{L}_{\text{appr}}(\mathcal{G}, \alpha + \Delta\alpha), \tag{3}$$

*where $\Delta\alpha$ is a perturbation term that satisfies $\Delta\alpha \geqslant 0$ and $\alpha + \Delta\alpha \in (0,1)$.*

Through this operation, the directed graph structure and the sparsity of the Laplacian matrix are maintained, which means there is no training burden to the subsequent GNN-based encoder. However, using this operation for data augmentation is very time-consuming, since the time complexity of computing Eq. (2) is $\mathcal{O}\left(n^2\right)$. To deal with it, we design an accelerating algorithm for computing this approximate eigenvector based on the power method [24].

We take advantage of the sparsity of the transition matrix $\tilde{\mathbf{P}}$ to compute the approximate eigenvector $\pi_{\text{appr}}$ and rewrite Eq. (2) to the form of discrete-time Markiv chain at time $t$ as

$$\pi_{\text{appr}}^{t+1} = (1-\alpha)\pi_{\text{appr}}^t\tilde{\mathbf{P}} + \frac{1}{n}\frac{\alpha}{1+\alpha}\mathbf{1}^{1\times n}. \tag{4}$$

As stated in Tong *et al.* [43], $\pi_{\text{appr}}^t$ has a special property that $\pi_{\text{appr}}^t\mathbf{1}^{n\times 1} = \frac{1}{1+\alpha}$. We then multiply the last term of Eq. (4) with $(1+\alpha)\pi_{\text{appr}}^t\mathbf{1}^{n\times 1}$ to

$$\pi_{\text{appr}}^{t+1} = (1-\alpha)\pi_{\text{appr}}^t\tilde{\mathbf{P}} + \frac{1}{n}\frac{\alpha}{1+\alpha}[(1+\alpha)\pi_{\text{appr}}^t\mathbf{1}^{n\times 1}]\mathbf{1}^{1\times n} = (1-\alpha)\pi_{\text{appr}}^t\tilde{\mathbf{P}} + \frac{\alpha}{n}\pi_{\text{appr}}^t\mathbf{1}^{n\times n}. \tag{5}$$

We can solve Eq. (5) iteratively using the power method [24]. Therefore, the complexity decreases to $\mathcal{O}\left(nk\right)$, where $k$ is the number of iteration times. Due to page limit, we do not discuss the number of iterations and convergence rate here. Please refer to [22] for more details. We take $k = 100$ and the tolerance as $1e-6$ throughout. It can be seen easily that for large-scale graphs, our method can significantly improve the speed and makes it possible to do Laplacian perturbation during the training, which is the basis for the framework proposed in Section 3. In addition, we compare the running time of our fast Laplacian perturbation with different data augmentation methods in Section 5.2.

---

[1]Note that adding self-loops to transform $\mathbf{P}$ into $\tilde{\mathbf{P}}$ is a common trick [17, 56], and we ignore the effect of this operation on the directed graph structure.

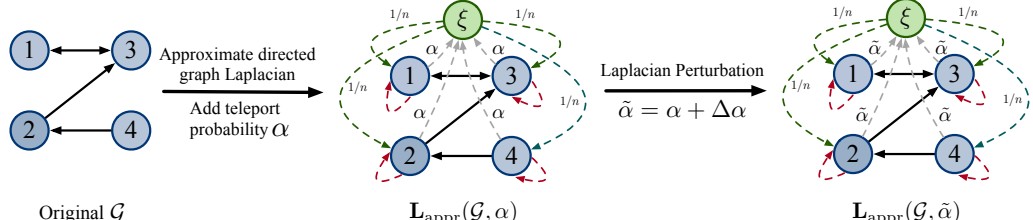

Figure 1: Illustration of Laplacian perturbation. $\xi$ is the auxiliary node defined in [43] and the red dotted lines represent adding the self-loops.

## 2.3 Justification of Laplacian Perturbation

Noticing the inherent connections between graph Laplacian and von Neumann entropy [62], we can use the proprieties of von Neumann entropy to analyze the Laplacian perturbation operation quantitatively. We start with defining the von Neumann entropy of directed graphs.

**DEFINITION 2** *Directed graph von Neumann entropy. Given a directed graph $\mathcal{G} = (\mathcal{V}, \mathcal{E})$, its von Neumann entropy [62] based on the $\mathbf{L}_{\mathrm{appr}}$ is defined as*

$$\tilde{\mathcal{H}}_{\mathrm{VN}}(\mathcal{G}, \alpha) = 1 - \frac{1}{n} - \frac{1}{2n^2} \left\{ \sum_{(u,v) \in \mathcal{E}} \left( \frac{1}{d_u^{out} d_v^{out}} + \frac{\pi_{\mathrm{appr}}(u)}{\pi_{\mathrm{appr}}(v) d_u^{out2}} \right) - \sum_{(u,v) \in \tilde{\mathcal{E}}} \frac{1}{d_u^{out} d_v^{out}} \right\}, \quad (6)$$

*where $\tilde{\mathcal{E}} = \{(u,v) \mid (u,v) \in \mathcal{E} \text{ and } (v,u) \notin \mathcal{E}\}$. $d_u^{in} = \sum_{v \in \mathcal{V}} \mathbf{A}(v,u)$ and $d_u^{out} = \sum_{v \in \mathcal{V}} \mathbf{A}(u,v)$ are the in- and out-degree of the node $u$.*

Since the Laplacian perturbation does not involve the nodes/edges and only changes the teleport probability connecting to the auxiliary node $\xi$ as shown in Figure 1, the number of nodes $n$ and the degrees of nodes $d_u$ (edge distribution) will not change. Conversely, the approximate eigenvector $\pi_{\mathrm{appr}}$ has been changed. According to Eq. 6, the essence of this operation is a perturbation to the directed graph entropy. We further introduce the perturbation error to quantify this impact.

**DEFINITION 3** *Perturbation error. Given the perturbation term $\Delta\alpha$, we define the perturbation error of directed graph von Neumann entropy caused by Laplacian perturbation as $\Delta\tilde{\mathcal{H}}_{\mathrm{VN}}(\alpha, \alpha + \Delta\alpha) = \tilde{\mathcal{H}}_{\mathrm{VN}}(\mathcal{G}, \alpha) - \tilde{\mathcal{H}}_{\mathrm{VN}}(\mathcal{G}, \alpha + \Delta\alpha)$.*

**THEOREM 1** *Monotonicity of the perturbation error. The perturbation error $\Delta\tilde{\mathcal{H}}_{\mathrm{VN}}$ increases monotonically with the Laplacian perturbation term $\Delta\alpha$.*

**THEOREM 2** *Bounds on the perturbation error. Given a directed graph $\mathcal{G} = (\mathcal{V}, \mathcal{E})$ and the teleport probability $\alpha$, the inequality*

$$0 < \Delta\tilde{\mathcal{H}}_{\mathrm{VN}}(\alpha, \alpha + \Delta\alpha) < \frac{1}{2n^2} \left\{ \sum_{(u,v) \in \mathcal{E}} \left( \frac{1}{d_u^{out2}} - \frac{\pi_{\mathrm{appr}}^{\alpha}(u)}{\pi_{\mathrm{appr}}^{\alpha}(v) d_u^{out2}} \right) \right\} \quad (7)$$

*holds. When the perturbation term $\Delta\alpha = 0$, $\Delta\tilde{\mathcal{H}}_{\mathrm{VN}} = 0$ and when $\Delta\alpha \to 1 - \alpha$, the perturbation error $\Delta\tilde{\mathcal{H}}_{\mathrm{VN}}$ towards the upper bound.*

We show in THEOREM 1 that Laplacian perturbation can provide contrastive information in various magnitude for the encoder to learn the representations by changing $\Delta\alpha$. Meanwhile, since it does not need to alter the structure and the number of nodes $n$ is generally high, the perturbation error will be in a very small range as shown in THEOREM 2. This can help the encoder to focus more on the directed graph structure rather than just learning from the errors. Proofs of THEOREM 1 and 2 are attached in the Supplementary Material.

## 3 Directed Graph Contrastive Learning

In this section, we introduce a new directed graph contrastive learning framework (DiGCL) with a more generalized objective, and then we present multi-task curriculum learning scheme to help DiGCL progressively learn from multiple easy-to-difficult contrastive views. The model illustration is in Figure 2 and the pseudo-code is given in the Supplementary Material.

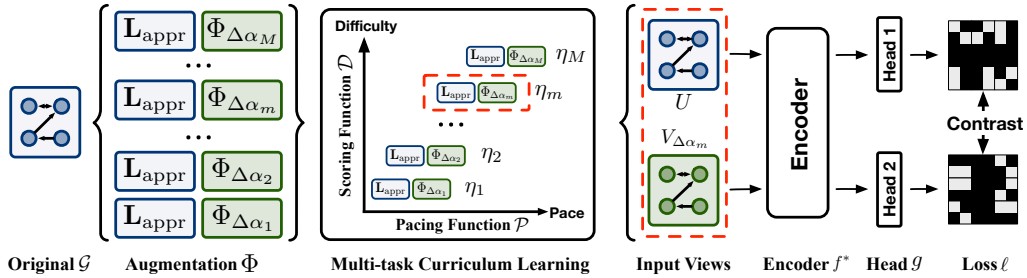

Figure 2: Illustration of our DiGCL model using Laplacian perturbation. For a directed graph, we first generate $M$ different pairs of contrastive views by Laplacian perturbation. The different contrastive view pairs are then scored by a scoring function and mapped to different training paces by a pacing function. Finally, the arranged contrastive view pairs are input into a shared encoder to progressively learn the unsupervised graph representation with contrastive loss.

## 3.1 Learning with Dynamic-view Contrastive Objective

As we have defined the directed graph data augmentation in Eq. (3), we follow the common graph contrastive learning (GCL) paradigm using our Laplacian perturbation.

**Fixed-view Objective.** GCL seeks to maximize the mutual information (MI) between the representations of augmented graphs under different views [63, 69]. Based on it, we design a framework for directed graph contrastive learning, which contains three steps. (1) First, given a directed graph $\mathcal{G} = (\mathcal{V}, \mathcal{E})$ and the teleport probability $\alpha$, we generate two correlated views using for the $\mathcal{G}$: $U_{\Delta\alpha_1} = \Phi_{\Delta\alpha_1}(\mathcal{G}, \alpha)$ and $V_{\Delta\alpha_2} = \Phi_{\Delta\alpha_2}(\mathcal{G}, \alpha)$ as a contrastive pair. This generation is through executing Laplacian perturbation $\Phi$ on the input $\mathcal{G}$ with the parameters $\Delta\alpha_1, \Delta\alpha_2$ separately. (2) Second, we take a GNN-based encoder $f(\cdot)$ to extract representation $\mathbf{H}_{\Delta\alpha_1} = f(U_{\Delta\alpha_1})$, $\mathbf{H}_{\Delta\alpha_2} = f(V_{\Delta\alpha_2})$ of two contrastive views. (3) Finally, the encoder is trained with the contrastive objective as

$$\max_f I(\mathbf{H}_{\Delta\alpha_1}; \mathbf{H}_{\Delta\alpha_2}) = \max_f I\left(f(\Phi_{\Delta\alpha_1}(\mathcal{G}, \alpha)); f(\Phi_{\Delta\alpha_2}(\mathcal{G}, \alpha))\right), \quad (8)$$

where $I(\cdot)$ is the mutual information. This framework lets the encoder $f(\cdot)$ learn the representation of two different views $U_{\Delta\alpha_1}, V_{\Delta\alpha_2}$ while preserving as much MI as possible.

As shown in Eq. (8), the view generation is controlled by two perturbation terms $\Delta\alpha_1$ and $\Delta\alpha_2$, which may affect the objective function. The current GCLs manually select these parameters in advance, *e.g.,* grid search, and fix contrastive views during training. This parameter selection strategy could make the encoder obtained by Eq. (8) learn only the contrastive information in some particular views and lead to a reduced generalization of the model [59].

**Dynamic-view Objective.** To deal with it, we propose a more generalized objective to learn from dynamic changing contrastive views. We rewrite Eq. (8) as

$$\max_{f^*} I(\mathbf{H}_{\Delta\alpha_1}; \mathbf{H}_{\Delta\alpha_2}) = \max_{f^*} \mathop{\mathbb{E}}_{\Delta\alpha_1, \Delta\alpha_2} I\left(f^*(\Phi_{\Delta\alpha_1}(\mathcal{G}, \alpha)); f^*(\Phi_{\Delta\alpha_2}(\mathcal{G}, \alpha))\right), \quad (9)$$

which requires the obtained optimal encoder $f^*$ works well with $\forall \Delta\alpha_1, \Delta\alpha_2 \in [0, 1 - \alpha)$ to learn a balanced and generalized representation. Note that *InfoMin* [42] designs a similar strategy that iteratively maximizes the MI in the worst views. In contrast, our objective maximizes the average MI in all views because finding the worst views is related to the downstream task, about which we have zero prior knowledge. Besides, unlike other contrastive learning methods that also use multiple contrastive views [41, 13, 61], there is no fixed number of views in our approach, and we do not preset the parameters to fix the contrastive views but let them dynamically change during training.

However, maximizing Eq. (9) is not easy. Varying the perturbation term brings changes to the MI of views, which may result in large variance during the training process and unable to converge [42]. To increase the stability, we empirically make one of the views unperturbed in this paper, *i.e.,* $\Delta\alpha_1 = 0$. We mark this view as $U$ and design a multi-task training scheme to optimize proposed objective[2].

## 3.2 Training using Multi-task Curriculum Learning

To tackle this problem, we propose a training scheme using curriculum learning, which utilizes *prior knowledge about the difficulty of the learning tasks* to learn from easy-to-difficult contrastive views.

---

[2]The objective of fixing a view is not exactly the same as the original one, and we ignore this trick's impact.

**Multi-task Curriculum Learning.** Curriculum Learning facilitates the optimization on such a complex objective by scheduling the sub-objectives in a certain order [2, 33]. Inspired by it, we regard the process of training DiGCL as a multi-task problem [33, 36, 10, 11, 26, 20] and decompose it into $M$ sub-tasks $\eta_1, \ldots, \eta_M$. Thus, the objective Eq. (9) can be rewritten as

$$\max_{f^*} \frac{1}{M} \sum_{m=1}^{M} \mathbb{E}_{\Delta\alpha_m} I\big(f^*(\mathbf{L}_{\text{appr}}(\mathcal{G}, \alpha)); f^*(\Phi_{\Delta\alpha_m}(\mathcal{G}, \alpha))\big), \tag{10}$$

where $\Delta\alpha_m \in \{\Theta_{1,\ldots,M}\}$ is a uniform partition of $\Theta = [0, 1-\alpha)$ and the sub-task $\eta_m$ is associated with the $\Delta\alpha_m$. Then we consider sub-tasks that have less difficulty to learn in the optimization process as **easy** sub-tasks and vice versa as **difficult** sub-tasks. We quantify this difficulty into *difficulty score* by means of a scoring function $\mathcal{D}(\cdot)$. Based on it, we learn sub-tasks sequentially in a certain order decided by the pacing function $\mathcal{P}(\cdot)$ to solve the main multi-task problem. Via information transmission by a shared encoder, the previously solved easy sub-tasks will assist in solving the next difficult sub-tasks, while the latter can fine-tune the encoder learned by the former.

**Scoring Function.** The scoring function $\mathcal{D}(\cdot)$ measures how difficult the sub-task $\eta_m$ is, and can be any function $\mathcal{D}(\cdot): \{\eta_1, \ldots, \eta_M\} \to \{d_1, \ldots, d_M\}$, where $d$ denotes the difficulty score. In each sub-task, there exists a pair of contrastive views as defined in Eq. (10). We empirically assume that the difficulty $d_m$ of the sub-task $\eta_m$ depend on how difficult it is for the encoder $f(\cdot)$ to learn the contrastive information from this sub-task. Referring to DEFINITION 3, we can find Laplacaian perturbation provides contrastive information to the encoder, and the perturbation term $\Delta\alpha$ controls how much of this information is. Therefore, we can define the scoring function $\mathcal{D}(\cdot)$ to calculate the difficulty $d_m$ of a sub-task $\eta_m$ as

$$d_m = \mathcal{D}(\eta_m) = 1 - \frac{\Delta\alpha_m}{1-\alpha}. \tag{11}$$

The higher this score represents the smaller the perturbation error, the less contrastive information provided, and the less the encoder is able to distinguish the difference between the two views, resulting in a more difficult sub-task. This idea coincides with the design of the discriminator in GAN [65, 9].

**Pacing Function.** The pacing function decides the learning sequence of sub-tasks and can be any function $\mathcal{P}(\cdot): \{1, \ldots, L\} \to \{d_1, \ldots, d_M\}$, where $L$ denotes all the learning iterations. We consider three function families [58]: logarithmic, exponential, and linear. Table in Figure 3(left) illustrates the pacing functions used in our experiments, which are parameterized by $(d_a, d_b)$. Here $d_a$ is the initial difficulty and $d_b$ is the difficulty at the end of training, thus any pacing function with $d_a = d_b$ is equivalent to fixed view contrastive learning. We mark their corresponding perturbation terms as $(\Delta\alpha_a, \Delta\alpha_b)$. Considering the score in Eq. (11) is defined in a continuous domain, we set the step size of the pacing function to 1, the finest-grained unit, *i.e.*, $L = M$. This can help the model to switch between different training views in a more delicate way.

| Name | Expression $\mathcal{P}_{(d_a, d_b)}(l)$ |
|------|------------------------------------------|
| log | $d_a + (d_b - d_a)\big(1 + 1/3 \log\big(\frac{l}{L} + e^{-3}\big)\big)$ |
| exp | $d_a + \frac{d_b - d_a}{e^3 - 1}\big(\exp\big(\frac{3l}{L}\big) - 1\big)$ |
| linear | $d_a + (d_b - d_a)\frac{l}{L}$ |

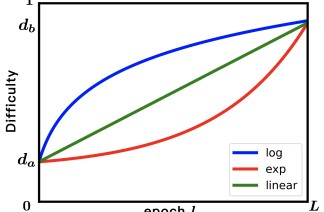

Figure 3: (left) pacing function definitions for the three families of pacing functions used throughout; (right) the plot of pacing function curves from each family. $l$ is the training epoch.

It is desired to find the optimal solution over the entire definition domain of $\Delta\alpha \in [0, 1-\alpha)$, we set the start point $\Delta\alpha_a = 0.9 - \alpha$ (0.9 is used because the boundary value cannot be obtained) and ending point $\Delta\alpha_b = 0$ in this paper, *i.e.*, $d_a = 1/9, d_b = 1$. More analysis on the pacing functions can be seen in the Supplementary Material.

**Contrastive loss.** After discussing how to measure expectations in the objective, our attention shifts to maximizing the MI. Several works have investigated the lower bound of MI in contrastive learning, here we adopt *InfoNCE* [28]. For $i^{\text{th}}$ node $u_i$ in view $U$, the node-wise objective is defined as

$$\ell(u_i, v_i) = -\log \frac{\exp\big(\mathcal{S}\big(\boldsymbol{z}_U^i, \boldsymbol{z}_{V_{\Delta\alpha_m}}^i\big)/\tau\big)}{\sum_{j=1}^{n} \exp\big(\mathcal{S}\big(\boldsymbol{z}_U^i, \boldsymbol{z}_{V_{\Delta\alpha_m}}^j\big)/\tau\big)}, \tag{12}$$

where $v_i$ is its corresponding positive node in view $V_{\Delta \alpha_m}$ and other nodes in view $V_{\Delta \alpha_m}$ is negative nodes of $u_i$. $z_U^i$ is the projection of node feature that $z_U^i \in \mathbf{Z}_U = g(\mathbf{H}_U)$. A non-linear transformation $g(\cdot)$ named projection head maps augmented representations $\mathbf{H}$ to another lower dimension where the contrastive loss is calculated. $\mathcal{S}(\cdot)$ denotes cosine similarity function and $\tau$ denotes the temperature parameter. The final loss is computed across all positive node pairs in the views of one pace. Note that GRACE [68] proposes similar *InfoNCE-like* loss function. Different from it, we do not treat intra-view nodes (nodes in $U$ except $u_i$) as the negative samples since GNN-based encoder is already able to learn intra-graph structure well, we want to focus on inter-graph contrastive information.

# 4 Related Work

**Directed Graph Neural Networks.** Several works have tried to learn the complex directed graph structure by defining motifs [25], using magnetic Laplacian [64], inheritance relationships [16], re-defining the propagation scheme from Markov process view [43, 23] and second-order proximity[44]. However, these methods are supervised, and they cannot learn the structural features without labels. Our model is unsupervised, which is the most significant difference between them.

**Directed Graph Data Augmentation.** Good data augmentations are the prerequisite for contrastive learning [63, 48, 54]. Most current data augmentation methods, such as dropping nodes or edges [63, 68, 69, 51] and attribute masking [15, 63, 55] are migrated from methods of visual representation [57, 66] with graph structural improvements. Other methods learn graph structure features by combining with traditional graph algorithms to go self-supervised, for example, subgraph sampling [63, 34, 52], graph diffusion based on PageRank/Heat kernel [13], and multi-hop neighbor prediction [31]. However, most of these methods are not optimized for directed graphs and thus cannot efficiently obtain structure information that can be used for self-supervised learning. The performance of existing methods on directed graphs can be seen in Table 2.

**Dynamic-view GCL and Multi-task Curriculum Learning.** Previous works [41, 13, 61] have found the powerful capabilities of using multiple contrastive views at the same time in contrastive learning. However, their methods require to set contrastive views in advance, which means the views are fixed during training [13, 41]. Unlike them, we do not preset the parameters to fix the contrastive views but let them dynamically change during training. However, our dynamic-view contrastive objective is hard to optimize, thus we adopt the idea of multi-task curriculum learning [33, 36, 10, 11, 26, 20] to progressively learn from multiple easy-to-difficult contrastive views. Besides, using curriculum learning [8, 53] in graph learning tasks is not a new idea, but we are the first to use curriculum learning as a solution to the self-supervised graph contrastive learning problem.

# 5 Experiments

We conduct extensive experiments to evaluate the effectiveness of our model. Our implement can be obtained at `https://github.com/flyingtango/DiGCL`. The Supplementary Material reports further details on the experiments and reproducibility.

## 5.1 Experimental Settings

**Experimental Task.** As explained in Section 2.1, our approach is more applicable to node-level contrastive learning, thus, we adopt the task of **node classification in directed graphs** [43] to verify the learning ability of models. Compared with the common experiments for undirected graphs [17, 56, 18], the challenge is that the given adjacency matrix $\mathbf{A}$ is asymmetric, which means message passing has its direction. We will also use datasets of undirected graphs for controlled trials. Task definition, model structures and configurations are included in the Supplementary Material.

**Datasets and Splitting.** We use several widely-used datasets including directed graph datasets: CORA-ML [3], CITESEER [37] and AM-PHOTO [38]; undirected graph datasets: PUBMED [27] and DBLP [30]. The split of the datasets would have a significant effect on models' results [38]. Thus, we divide the datasets randomly and conduct multiple tests to achieve consistent outcomes. For train/validation/test split, following the rules in GCN [17], we choose 20 labels per class for training set, 500 labels for validation set, and rest for the test set.

**Baselines.** We compare our model to the 11 SOTA models divided into four main categories: 1) **supervised** models for undirected graph with GCN [17], GAT [46] and [18]; 2) **supervised** models for directed graph with MagNet [64] and DiGCN [43]; 3) **self-supervised** models without augmentations including DGI [47] and GMI [32]; 4) **contrastive learning** models with augmentations containing MVGRL [13], GraphCL [63], GRACE [68], and GCA [69].

## 5.2 Experimental Results

**Overall accuracy.** The performance comparisons between our model and baselines on five datasets are reported in Table 1. We use a two-layer GCN as our encoder and first train our model in an unsupervised manner to obtain the embedding. Then we take a simple $\ell_2$-regularized logistic regression as the classifier [47]. For curriculum learning scheme, we select the log pacing function and the start and ending difficulties are set in Section 3.2. We train all models according to their default settings, then calculate mean test accuracy with STD in percent (%) averaged over 20 random dataset splits with random weight initialization.

It can be seen easily that our methods achieve the state-of-the-art results on all datasets. In general, the unsupervised methods including MVGRL, GRACE, and GCA, do not perform well on directed graphs compared to their good performance in undirected graphs. This is mainly due to the models' data augmentation methods are not applicable to digaphs, resulting in the inability to learn contrastive information from complex directed structures. Notice that GraphCL performs relatively mediocre in unsupervised methods, most notably because it focuses mainly on graph-level unsupervised methods, and does not apply well to node-level contrastive learning task. Since DGI and GMI do not require data augmentation to provide contrastive information, they perform very well on both undirected and directed graphs, which shows good suitability for different graph structure. Moreover, supervised methods such as GCN, GAT and APPNP are inferior to DiGCN and MagNet, which are specifically designed for directed graphs, in terms of performance. Since our Laplacian perturbation uses the approximate Laplacian matrix proposed by DiGCN, we compare their performance. It is not difficult to find that our model outperforms DiGCN on all datasets, which shows that contrastive learning can learn good encoders by performing a certain data augmentation in an unsupervised manner.

Table 1: Overall accuracy (%) with STD. "No Curr" means do not use curriculum learning and we set two fixed views as $\Delta\alpha_a = \Delta\alpha_b = 1 - \alpha$. "Random" means random order, "Anti Curr" means using anti-curriculum order and "Curr" indicates using curriculum order. The best results are highlighted with **bold** and the second are marked with underline. OOM means out of memory on a 12GB GPU.

| | Method | DIRECTED | | | UNDIRECTED | |
| --- | --- | --- | --- | --- | --- | --- |
| | | CORA-ML | CITESEER | AM-PHOTO | PUBMED | DBLP |
| SUPERVISED | GCN [17] | 70.92 ± 0.39 | 63.00 ± 0.45 | 88.52 ± 0.47 | 78.78 ± 0.30 | 73.54 ± 0.77 |
| | GAT [46] | 72.22 ± 0.57 | 63.73 ± 0.57 | 88.36 ± 1.25 | 77.49 ± 0.47 | 76.08 ± 0.54 |
| | APPNP [18] | 70.31 ± 0.67 | 61.63 ± 0.63 | 87.43 ± 0.98 | 79.35 ± 0.48 | 77.92 ± 0.75 |
| | MagNet [64] | 76.32 ± 0.10 | 65.04 ± 0.47 | 86.80 ± 0.65 | 74.23 ± 0.46 | 69.73 ± 0.98 |
| | DiGCN [43] | 77.03 ± 0.70 | 64.60 ± 0.60 | 88.66 ± 0.51 | 76.79 ± 0.49 | 73.37 ± 0.72 |
| UNSUPERVISED | DGI[47] | 75.21 ± 1.29 | 64.58 ± 1.78 | 85.25 ± 0.59 | 74.11 ± 0.62 | 76.53 ± 1.24 |
| | GMI[32] | 76.59 ± 0.35 | 63.29 ± 0.70 | 81.12 ± 0.01 | 80.27 ± 0.16 | 76.66 ± 0.48 |
| | MVGRL[13] | 76.67 ± 0.12 | 62.22 ± 0.02 | 86.15 ± 0.21 | 79.98 ± 0.04 | OOM |
| | GraphCL [63] | 67.34 ± 0.12 | 57.84 ± 0.11 | 67.66 ± 0.05 | 75.29 ± 0.08 | 77.85 ± 0.22 |
| | GRACE [68] | 73.88 ± 0.25 | 61.20 ± 0.20 | 87.95 ± 0.32 | 79.54 ± 0.05 | 78.03 ± 0.09 |
| | GCA [69] | 76.32 ± 0.33 | 63.25 ± 0.10 | 87.35 ± 0.27 | 79.81 ± 0.61 | 77.83 ± 0.35 |
| | **Ours** + No Curr | 75.86 ± 0.09 | 66.99 ± 0.54 | 87.32 ± 0.14 | 79.57 ± 0.12 | 78.28 ± 0.05 |
| | **Ours** + Random | 76.52 ± 1.66 | 67.15 ± 0.82 | 89.03 ± 0.46 | **80.75 ± 0.10** | 79.58 ± 0.14 |
| | **Ours** + Anti Curr | 76.12 ± 1.04 | 66.83 ± 1.13 | 88.83 ± 0.73 | 80.22 ± 0.37 | 79.42 ± 0.15 |
| | **Ours** + Curr | **77.53 ± 0.14** | **67.42 ± 0.14** | **89.41 ± 0.11** | 80.69 ± 0.08 | **79.70 ± 0.13** |

**Ablation study on curriculum learning**. We validate the effectiveness of curriculum learning strategy and the results are shown in Table 1. We find that even with the contrastive views containing the maximum information (with the largest perturbation term), the effect of the model without curriculum learning is significantly lower than the model with curriculum learning. This means

that features learned from fixed views is always incomplete. Besides, we also find that the order of curriculum learning is vital, the ACC of learning from easy-to-difficult > random order > difficult-to-easy. This substantially validates that scoring function do help to learn from multiply tasks.

**Impact of different pacing functions.** We study the three different pacing functions defined in Table 3 and the results in Figure 4(a) show that using different path functions can have an impact on the performance but not as much as changing the order of curriculum learning. Among them, the log pacing function performs best as it speeds up learning on easy tasks and stays on harder tasks for more epochs, thus helping the model to grasp the more subtle differences between contrastive views.

**Effect of Laplacian perturbation and other data augmentation methods.** Table 2 shows the effect of the different data augmentation methods on the five contrastive learning models. We our Laplacian perturbation works best on both directed and undirected graphs. But the difference with other methods is not significant on the undirected graph dataset PUBMED. In addition to this, we also compare with random removing edges used in GRACE under our DiGCL framework. We find that the data augmentation approach without damaging the graph structure can achieve better performance.

Table 2: Overall accuracy (%) with STD with various data augmentation methods. Our model uses curriculum learning, and use the log pacing function. The best results are highlighted with **bold**.

| Methods | Data Augmentation Method | CITESEER | AM-PHOTO | PUBMED |
|---|---|---|---|---|
| GraphCL [63] | Edge perturbation | $57.84 \pm 0.11$ | $67.66 \pm 0.05$ | $75.29 \pm 0.09$ |
| | Node dropping | $57.45 \pm 0.12$ | $66.69 \pm 0.07$ | $75.25 \pm 0.08$ |
| | Subgraph sampling | $57.59 \pm 0.10$ | $66.75 \pm 0.07$ | $74.75 \pm 0.11$ |
| GRACE [68] | Random removing edges | $61.20 \pm 0.20$ | $87.95 \pm 0.32$ | $79.54 \pm 0.05$ |
| MVGRL [13] | Graph diffusion with heat kernel | $61.22 \pm 0.07$ | $79.63 \pm 0.31$ | $78.54 \pm 0.33$ |
| | Graph diffusion with PageRank kernel | $62.22 \pm 0.02$ | $86.15 \pm 0.21$ | $79.98 \pm 0.04$ |
| GCA [69] | Removing edges by degree score | $63.25 \pm 0.10$ | $87.35 \pm 0.27$ | $79.81 \pm 0.61$ |
| | Removing edges by PageRank score | $62.21 \pm 0.16$ | $86.88 \pm 0.37$ | $78.92 \pm 0.37$ |
| | Removing edges by eigenvalue score | $63.12 \pm 0.08$ | $87.02 \pm 0.56$ | $79.70 \pm 0.09$ |
| Ours | Random removing edges | $64.97 \pm 0.08$ | $88.45 \pm 0.01$ | $79.66 \pm 0.13$ |
| | Laplacian perturbation | $\mathbf{67.42 \pm 0.14}$ | $\mathbf{89.41 \pm 0.11}$ | $\mathbf{80.69 \pm 0.08}$ |

**Generalization of Laplacian perturbation.** Here, we will add generalizability experiments that migrate Laplacian perturbation to MVGRL [13] and GCA [69]. We replace the topological data augmentation method in the original model with our Laplacian perturbation and follow all the configurations in the original model, including the parameters of the data augmentation, the number of contrastive views, the structure of the model, and the training parameters. We test the generalization performance of our Laplacian perturbation in the node classification task. The results are shown in Figure 4(b). We can clearly find that the performance of the model on directed graphs can be improved after using Laplacian perturbation, which indicates that our method can help existing contrastive learning models to learn structure information from complex networks.

**Monotonicity of the perturbation error in real-world datasets.** To visualize the variation of the perturbation error with the perturbation term, we empirically show the variation of directed graph entropy in the three real-world directed graph datasets: CORA-ML, CITESEER and AM-PHOTO. We

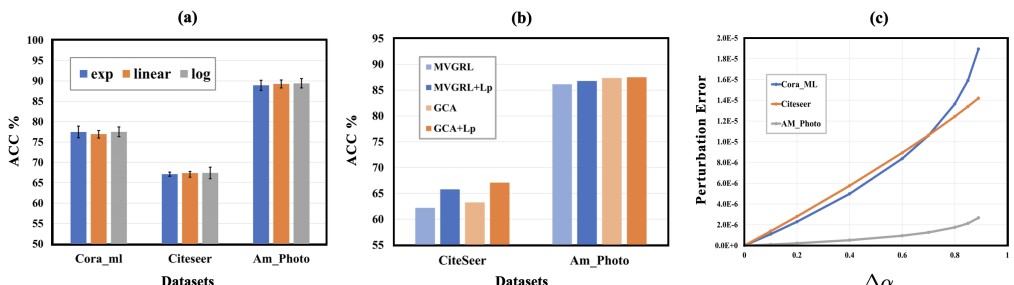

Figure 4: (a) node classification results with various pacing functions; (b) node classification results for the generalizability of Laplacian perturbation on two datasets, Lp means with Laplacian perturbation; (c) perturbation error with various perturbation terms $\Delta\alpha$.

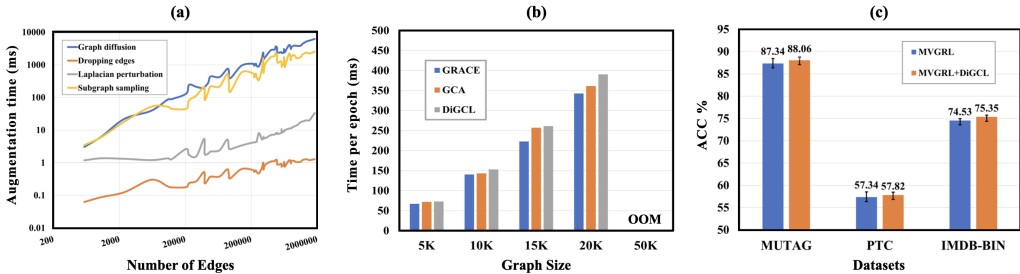

Figure 5: (a) time of different data augmentation methods on random graphs with different size; (b) running time per epoch of different models on random size graphs, OOM means out of memory; (c) generalization experiment of graph classification task based on MVGRL.

set $\alpha = 0.1$ and take $\Delta\alpha \in \{0, 0.1, 0.2, 0.4, 0.6, 0.7, 0.8, 0.85, 0.89\}$. The results are shown in Figure 4(c), and it is clearly shown that as the perturbation term increases, the degree of perturbation is elevated, and the perturbation error will increase.

**Augmentation time at different graph scales.** We construct 20 random directed graphs of different scales, with the number of edges ranging from 1K to 2M, then test four different data augmentation methods' mean running time in milliseconds (ms) for 5 times. Figure 5(a) summarizes the results and shows that our Laplacian perturbation is very competitive in terms of runtime among these methods. In contrast to the graph diffusion, which also require the computation of eigenvectors, we can reduce the time by a factor of one hundred with our fast algorithm proposed in Section 2.2. This makes it possible to dynamically perform Laplacian perturbation to generate contrastive views during training.

**Running time with different graph size.** To generate arbitrary graph size, we construct a simple random graph with $N$ nodes and assign $10N$ directed edges uniformly at random. We compare with GRACE[68] and GCA [69] since these two models have the similar accuracy with ours. We record the running time of the models for each epoch at different graph sizes, and the results are shown in Figure 5(b). We can find that processing 20,000 nodes and 200,000 edges with 12 GB memory is the limit for these three models. Since our model requires a Laplacian perturbation operation, this will take more time than the data augmentation performed by GCA and GRACE.

**Generalization to graph classification task.** To validate the generalization ability of our DiGCL model, we migrate it to MVGRL [13] and test on three graph classification datasets: MUTAG [19], PTC [19], and IMDB-BIN [60]. The generalized MVGRL model is denoted by MVGRL+DiGCL and the results are shown in Figure 5(c). Using our model can improve the accuracy on all datasets, however, the improvement is not significant. We consider there are three main potential reasons for this: 1) the datasets are undirected graphs, and the effect of using Laplacian perturbation on them is not as obvious as in directed graphs; 2) the graph classification problem requires more graph-level contrastive information, while our model focuses on node-level; 3) our scoring function designed for node-level curriculum learning is not suitable for graph-level difficulty measure.

## 6  Conclusion, Limitation and Future Work

In this paper, we design a directed graph data augmentation scheme called Laplace perturbation and theoretically investigate how it can provide contrastive information without changing the directed graph structure. Moreover, we present the DiGCL which utilizes Laplacian perturbation and curriculum learning to progressively learn from dynamic easy-to-difficult contrastive views. Finally, we use several tasks on various datasets to demonstrate the effectiveness and generalization ability of our proposed DiGCL. We empirically show that DiGCL can retain more structural features of directed graphs than other GCL models while providing adequate contrastive information. Extensive experiments show that our DiGCL outperforms the state-of-the-art approaches.

As elaborated in Section 2.1 and 5.2, our approach is more suitable for obtaining node-level contrastive information. This limits us to use the model to solve some graph-level problems, such as pre-training on protein molecular, drug property prediction, etc. We will extend our approach to graph-level tasks in subsequent work. Meanwhile, we manually design the pacing function to preplan the training speed, such as linear or exponential functions. However, in many cases, the pacing during training is not perfectly planned by the ideal curve. We need to dynamically adjust the pacing function according to the training situation, such as using self-paced learning [20].

## Acknowledgments and Disclosure of Funding

The authors would like to thank Changsheng Sun, Xu Cai and Jinyang Wang for the great help in discussion and proofreading. This research is supported by NUS Research Scholarship.

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
