# Supplementary Material for the Paper: Directed Graph Contrastive Learning

**Zekun Tong**[1]   **Yuxuan Liang**[1]   **Henghui Ding** [2,3,*]
**Yongxing Dai** [4]   **Xinke Li**[1]   **Changhu Wang**[2]
[1]National University of Singapore  [2]ByteDance  [3]ETH Zürich  [4]Peking University
{zekuntong,liangyuxuan,xinke.li}@u.nus.edu
henghui.ding@vision.ee.ethz.ch, yongxingdai@pku.edu.cn
changhu.wang@bytedance.com

## A  Proofs of Theorems

### A.1  Proof of Theorems 1

**THEOREM 1.** *Monotonicity of the perturbation error*. *The perturbation error $\Delta\tilde{\mathcal{H}}_{\mathrm{VN}}$ increases monotonically with the Laplacian perturbation term $\Delta\alpha$.*

*Proof.* The perturbation error is defined in DEFINITION (3) of the main text as

$$\Delta\tilde{\mathcal{H}}_{\mathrm{VN}}(\alpha, \alpha + \Delta\alpha) = \frac{1}{2n^2}\left\{\sum_{(u,v)\in\mathcal{E}}\left(\frac{\pi_{\mathrm{appr}}^{\alpha+\Delta\alpha}(u)}{\pi_{\mathrm{appr}}^{\alpha+\Delta\alpha}(v)d_u^{\mathrm{out}^2}} - \frac{\pi_{\mathrm{appr}}^{\alpha}(u)}{\pi_{\mathrm{appr}}^{\alpha}(v)d_u^{\mathrm{out}^2}}\right)\right\}. \tag{1}$$

We start out the proof from Eq. (1) in the main text, leading to

$$(1-\alpha)\pi_{\mathrm{appr}}^{\alpha}\tilde{\mathbf{P}} + \frac{1}{n}\frac{\alpha}{1+\alpha}\mathbf{1}^{1\times n} = \pi_{\mathrm{appr}}^{\alpha}, \tag{2}$$

and the approximate eigenvector component for node $u$ is

$$\pi_{\mathrm{appr}}^{\alpha}(u) = (1-\alpha)\sum_{i,(i,u)\in\mathcal{E}}\pi_{\mathrm{appr}}^{\alpha}(i)\tilde{\mathbf{P}}(i,u) + \frac{1}{n}\frac{\alpha}{1+\alpha}. \tag{3}$$

In the [19], they assume that the eigenvector component is proportional to the in-degree of the corresponding node when the neighborhood of this node has similar out-degree and in-degree, *i.e.*,

$$\frac{\sum_{i,(i,u)\in\mathcal{E}}\pi_{\mathrm{appr}}^{\alpha}(i)\tilde{\mathbf{P}}(i,u)}{\sum_{i,(i,v)\in\mathcal{E}}\pi_{\mathrm{appr}}^{\alpha}(i)\tilde{\mathbf{P}}(i,v)} \approx \frac{d_u^{\mathrm{in}}}{d_v^{\mathrm{in}}} = \frac{cd_u^{\mathrm{in}}}{cd_v^{\mathrm{in}}}, \tag{4}$$

where the constant $c$ controls the $d_u^{\mathrm{in}}$ and $\pi_{\mathrm{appr}}^{\alpha}(i)\tilde{\mathbf{P}}(i,u)$ at the same scale. Meanwhile, they experimentally verify that even under this strong assumption, the calculated von Neumann entropy does not show significant errors [19]. Thus, we adopt their assumption and simply Eq. (4) to

$$\sum_{i,(i,u)\in\mathcal{E}}\pi_{\mathrm{appr}}^{\alpha}(i)\tilde{\mathbf{P}}(i,u) \approx cd_u^{\mathrm{in}}. \tag{5}$$

---

*Corresponding author

35th Conference on Neural Information Processing Systems (NeurIPS 2021).

We further let $0 < \alpha_1 < \alpha_2 < 1$ and take them into Eq. (1)

$$
\begin{aligned}
\Delta\tilde{\mathcal{H}}_{\mathrm{VN}}(\alpha_1,\alpha_2) &= \frac{1}{2n^2}\left\{\sum_{(u,v)\in\mathcal{E}}\frac{1}{d_u^{\mathrm{out}2}}\left(\frac{(1-\alpha_2)cd_u^{\mathrm{in}}+\frac{1}{n}\frac{\alpha_2}{1+\alpha_2}}{(1-\alpha_2)cd_v^{\mathrm{in}}+\frac{1}{n}\frac{\alpha_2}{1+\alpha_2}}-\frac{(1-\alpha_1)cd_u^{\mathrm{in}}+\frac{1}{n}\frac{\alpha_1}{1+\alpha_1}}{(1-\alpha_1)cd_v^{\mathrm{in}}+\frac{1}{n}\frac{\alpha_1}{1+\alpha_1}}\right)\right\} \\
&= \frac{1}{2n^2}\left\{\sum_{(u,v)\in\mathcal{E}}\frac{1}{d_u^{\mathrm{out}2}}\left(\frac{n(1-\alpha_2{}^2)cd_u^{\mathrm{in}}+\alpha_2}{n(1-\alpha_2{}^2)cd_v^{\mathrm{in}}+\alpha_2}-\frac{n(1-\alpha_1{}^2)cd_u^{\mathrm{in}}+\alpha_1}{n(1-\alpha_1{}^2)cd_v^{\mathrm{in}}+\alpha_1}\right)\right\} \\
&= \frac{1}{2n^2}\left\{\sum_{(u,v)\in E}\frac{1}{d_u^{\mathrm{out}2}}\left(\frac{(n(1-\alpha_2{}^2)\alpha_1 c(d_u^{\mathrm{in}}-d_v^{\mathrm{in}})-n(1-\alpha_1{}^2)\alpha_2 c(d_u^{\mathrm{in}}-d_v^{\mathrm{in}}))}{(n(1-\alpha_2{}^2)cd_v^{\mathrm{in}}+\alpha_2)(n(1-\alpha_1{}^2)cd_v^{\mathrm{in}}+\alpha_1)}\right)\right\} \quad (6) \\
&= \frac{1}{2n^2}\left\{\sum_{(u,v)\in\mathcal{E}}\frac{d_u^{\mathrm{in}}-d_v^{\mathrm{in}}}{d_u^{\mathrm{out}2}}\left(\frac{nc(1-\alpha_2{}^2)\alpha_1-nc(1-\alpha_1{}^2)\alpha_2}{(n(1-\alpha_2{}^2)cd_v^{\mathrm{in}}+\alpha_2)(n(1-\alpha_1{}^2)cd_v^{\mathrm{in}}+\alpha_1)}\right)\right\} \\
&= \frac{1}{2n^2}\left\{\sum_{(u,v)\in\mathcal{E}}\frac{d_u^{\mathrm{in}}-d_v^{\mathrm{in}}}{d_u^{\mathrm{out}2}}\left(\frac{nc(1/\alpha_2-\alpha_2)-nc(1/\alpha_1-\alpha_1)}{(n(1-\alpha_2{}^2)cd_v^{\mathrm{in}}+\alpha_2)(n(1-\alpha_1{}^2)cd_v^{\mathrm{in}}+\alpha_1)/(\alpha_1\alpha_2)}\right)\right\}.
\end{aligned}
$$

Since $d_v^{\mathrm{in}} \leqslant n$ and $d_u^{\mathrm{out}} \leqslant n$,

$$
\begin{aligned}
\Delta\tilde{\mathcal{H}}_{\mathrm{VN}}(\alpha_1,\alpha_2) &\geqslant \frac{1}{2n^2}\sum_{(u,v)\in\mathcal{E}}(d_u^{\mathrm{in}}-d_v^{\mathrm{in}})\underbrace{\left(\frac{\frac{c}{n}(1/\alpha_2-\alpha_2-1/\alpha_1+\alpha_1)}{(n^2(1-\alpha_2{}^2)c+\alpha_2)(n^2(1-\alpha_1{}^2)c+\alpha_1)/(\alpha_1\alpha_2)}\right)}_{constant\ C} \\
&\geqslant \frac{C}{2n^2}\sum_{(u,v)\in\mathcal{E}}(d_u^{\mathrm{in}}-d_v^{\mathrm{in}}).
\end{aligned} \quad (7)
$$

As $0 < \alpha_1 < \alpha_2 < 1$, the constant $C < 0$. And the edges point from node $u$ to node $v$, thus the term $\sum_{(u,v)\in\mathcal{E}}(d_u^{\mathrm{in}}-d_v^{\mathrm{in}}) < 0$. Therefore,

$$
\Delta\tilde{\mathcal{H}}_{\mathrm{VN}}(\alpha_1,\alpha_2) > 0. \quad (8)
$$

Clearly, the perturbation error $\Delta\tilde{\mathcal{H}}_{\mathrm{VN}}$ increases monotonically with the Laplacian perturbation term $\Delta\alpha$. The proof is concluded. $\qquad\square$

## A.2 Proof of Theorems 2

**THEOREM 2.** ***Bounds on the perturbation error***. *Given a directed graph $\mathcal{G} = (\mathcal{V}, \mathcal{E})$ and the teleport probability $\alpha$, the inequality*

$$
0 < \Delta\tilde{\mathcal{H}}_{\mathrm{VN}}(\alpha,\alpha+\Delta\alpha) < \frac{1}{2n^2}\left\{\sum_{(u,v)\in\mathcal{E}}\left(\frac{1}{d_u^{\mathrm{out}2}}-\frac{\pi_{\mathrm{appr}}^{\alpha}(u)}{\pi_{\mathrm{appr}}^{\alpha}(v)d_u^{\mathrm{out}2}}\right)\right\} \quad (9)
$$

*holds. When the perturbation term $\Delta\alpha = 0$, $\Delta\tilde{\mathcal{H}}_{\mathrm{VN}} = 0$ and when $\Delta\alpha \to 1-\alpha$, the perturbation error $\Delta\tilde{\mathcal{H}}_{\mathrm{VN}}$ towards the upper bound.*

*Proof.* From THEOREM 1, $\Delta\tilde{\mathcal{H}}_{\mathrm{VN}}$ increases monotonically with the Laplacian perturbation term $\Delta\alpha$. Thus, when $\Delta\alpha = 0$, $\Delta\tilde{\mathcal{H}}_{\mathrm{VN}} = 0$. For the upper bound of the perturbation error, we start with Eq. (2) that

$$
(1-\alpha-\Delta\alpha)\pi_{\mathrm{appr}}^{\alpha+\Delta\alpha}\tilde{\mathbf{P}} + \frac{1}{n}\frac{\alpha+\Delta\alpha}{1+\alpha+\Delta\alpha}\mathbf{1}^{1\times n} = \pi_{\mathrm{appr}}^{\alpha+\Delta\alpha}. \quad (10)
$$

Since $\pi_{appr}$ is the stationary distribution and $\tilde{\mathbf{P}}$ is transition matrix, $\|\pi_{appr}\tilde{\mathbf{P}}\|_\infty \leqslant \|\pi_{appr}\|_\infty\|\tilde{\mathbf{P}}\|_\infty \leqslant 1$. It is easy to observe that when $\alpha+\Delta\alpha \to 1$, $\pi_{\mathrm{appr}}^{\alpha+\Delta\alpha} \to \frac{1}{2n}\mathbf{1}^{1\times n}$, which means $\pi_{\mathrm{appr}}^{\alpha+\Delta\alpha}(u), \pi_{\mathrm{appr}}^{\alpha+\Delta\alpha}(v)$ are equivalent as $\alpha + \Delta\alpha \to 1$. Thus,

$$
\Delta\tilde{\mathcal{H}}_{\mathrm{VN}}(\alpha,\alpha+\Delta\alpha) \to \frac{1}{2n^2}\left\{\sum_{(u,v)\in\mathcal{E}}\left(\frac{1}{d_u^{\mathrm{out}2}}-\frac{\pi_{\mathrm{appr}}^{\alpha}(u)}{\pi_{\mathrm{appr}}^{\alpha}(v)d_u^{\mathrm{out}2}}\right)\right\}, \quad (11)
$$

when $\alpha + \Delta\alpha \to 1$. The proof is concluded. $\qquad\square$

# B  Supplementary Experiments

We will show here the supplementary experiments which are not described in the main text.

## B.1  Experiments on the pacing function

Here, we provide more experiments on the pacing function. The pacing function determines in which order the contrastive views enter into the model. We want to know whether the information learned by the model in the easy contrastive view can help subsequent learning in the more difficult view.

**Accuracy with epoch for different pacing functions.** First, we give the results of the val accuracy changes with three different pacing functions in CORA-ML and AM-PHOTO in Figure 1(a) and 1(b) separately. We can find that different pacing functions perform differently at different training stages. Linear performs evenly throughout the training process; Exp improves faster at the beginning of training, but plateaus in the later stages; Log improves slowly at the beginning of training, but it continues to improve and achieves the best results at the end of training. The main reason is the log pacing function speeds up learning on easy tasks and stays on harder tasks for more epochs, helping the model to grasp the more subtle differences between contrastive views. This is the cause of its ability to consistently improve his performance in the later stages.

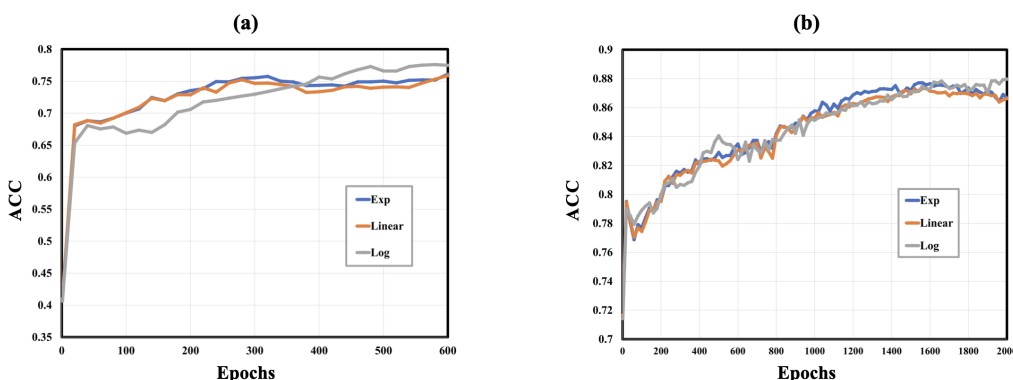

Figure 1: (a) performance of node classification task on CORA-ML with different pacing functions; (b) performance of node classification task on AM-PHOTO with different pacing functions.

**Sensitivity analysis for initial and ending difficulty.** Recalling the analysis in Section 3.2 of the main text, to obtain comprehensive contrastive information on the one hand, and to reduce the need for hyperparameters on the other hand, we set the initial perturbation term $\Delta\alpha_a$ to 0.8 and the ending perturbation term $\Delta\alpha_b$ to 0. In this experiment, we will explore the effect of different initial and ending difficulties on the accuracy of the model. We traverse the $\Delta\alpha_a, \Delta\alpha_b \in \{0, 0.1, 0.2, 0.3, 0.4, 0.5, 0.6, 0.7, 0.8\}$ and make sure $\Delta\alpha_a \geqslant \Delta\alpha_b$ ($\Delta\alpha_a = \Delta\alpha_b$ is equivalent to fixed view contrastive learning). We use two datasets and three pacing functions in the experiment. The results are shown in Figure 2. We can clearly find that setting the perturbation terms as the boundary values allows the model to learn all views as much as possible, thus improving the performance. Also, comparing the results of **Log**, **Liner**, and **Exp**, we can find that using log as the pacing function can get more stable and accurate results. This is consistent with the conclusion we obtain from the experiments in the main text.

From these experiments, we can draw a few empirical conclusions as follows.

- Log-based pacing function performs the best of the three pacing functions, but not too far from the other two pacing functions.

- The best results are obtained by setting the start and end points to be the boundary points of the Laplacian perturbation parameter space.

- The order in which the views are learned is crucial, with contrastive views working best from easy to difficult (Concluded from Table 1 in the main text).

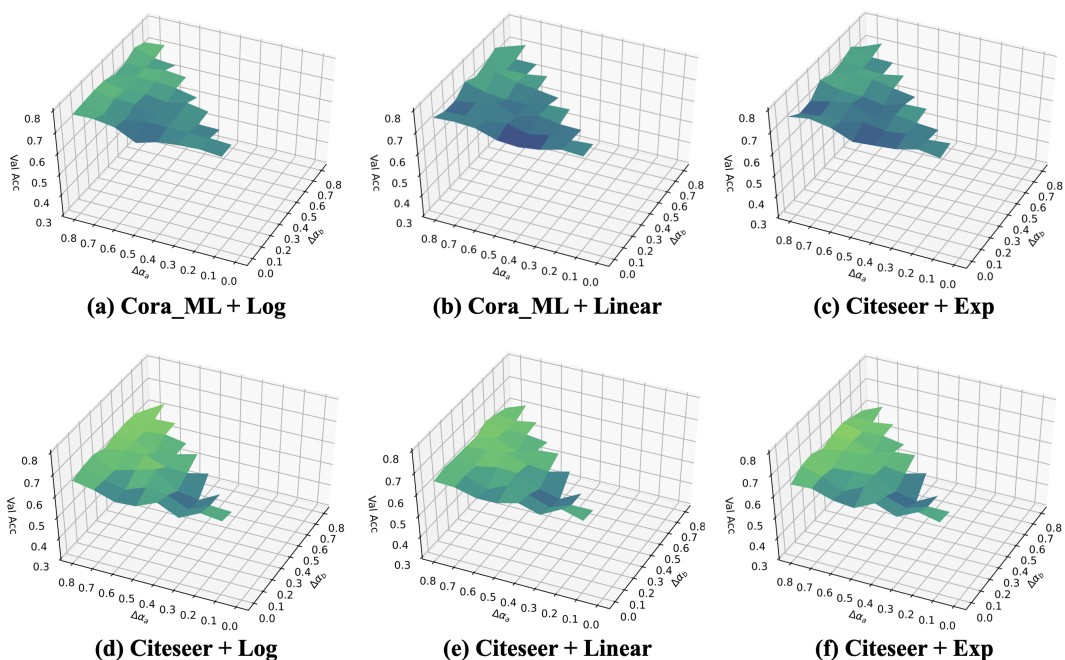

**Figure 2:** Validation accuracy of node classification task with different perturbation terms. The shade of the color represents the accuracy, with lighter shades indicating higher accuracy.

For the starting and ending difficulty scores, in accordance with the second conclusion, we consider that it is better to take the boundary values, which are effective and do not require parameter selection. For the type of pacing functions, according to the first and third conclusions, the different pacing functions have an impact on the results of the model but are not as important as the learning order. We believe that any pacing functions that satisfy the order of easy to difficult can be chosen.

## C Reproducibility Details

To support the reproducibility of the results, in this paper, we detail the task, datasets, the baseline setting, and pseudocodes. We implement the DiGCL and all baseline models using the python library of PyTorch [2], Pytorch-Geometric [2] and DGL [17]. All the experiments are conducted on a server with one 12GB GPU (NVIDIA TITAN V), two CPUs (Intel Xeon E5 × 2) and Ubuntu 18.04 System.

### C.1 Node Classification Task in Directed graphs

First, we define our task used in the main task as follow.

**DEFINITION 1.** *Directed Graph Node Classification. Given a directed graph $\mathcal{G} = (\mathcal{V}, \mathcal{E})$ with adjacency matrix $\mathbf{A}$, and node feature matrix $\mathbf{X} \in \mathbb{R}^{n \times c}$, where $n = |\mathcal{V}|$ is the number of nodes and $c$ is the feature dimension. Given a subset of nodes $\mathcal{V}_l \subset \mathcal{V}$, where nodes in $\mathcal{V}_l$ have observed labels and generally $|\mathcal{V}_l| << |\mathcal{V}|$. For semi-supervised (or supervised) methods, the task is using the labeled subset $\mathcal{V}_l$, node feature matrix $\mathbf{X}$ and adjacency matrix $\mathbf{A}$ predict the unknown label in $\mathcal{V}_{ul} = \mathcal{V} \smallsetminus \mathcal{V}_l$. For self-supervised methods, the task requires to use the adjacency matrix $\mathbf{A}$ and the node feature matrix $\mathbf{X}$ to learn node representation without labels.*

Specifically, after the model has unsupervisedly learned the node feature representation, simple classical classification algorithms, such as logistic regression, SVM, and etc., can be used to categorize the nodes from the node representation, which is a semi-supervised step. In this paper, all experiments in semi-supervised learning are set up the same, including the division of the datasets and the number of trial repetitions.

---

[2]https://pytorch.org

## C.2   Datasets Details

We use five open access datasets in the task of node classification. Label rate is the fraction of nodes in the training set per class. We use 20 labeled nodes per class to calculate the label rate.

Table 1: Datasets Details for Node Classification

| Datasets | Graph type | Nodes | Edges | Classes | Features | Label rate |
|---|---|---|---|---|---|---|
| CORA-ML [1] | Directed Graphs | 2995 | 8416 | 7 | 2879 | 4.67% |
| CITESEER [12] | Directed Graphs | 3312 | 4715 | 6 | 3703 | 3.62% |
| AM-PHOTO [13] | Directed Graphs | 7650 | 143663 | 8 | 745 | 2.10% |
| PUBMED [9] | Undirected Graphs | 18230 | 79612 | 3 | 500 | 0.33% |
| DBLP [10] | Undirected Graphs | 17716 | 105734 | 4 | 1639 | 0.45% |

Besides, to verify the generalizability of our approach, we also perform the graph classification task on three undirected graph datasets in the experiments. We use the following: MUTAG [8] containing mutagenic compounds, PTC [8] containing compounds tested for carcinogenicity, and IMDB-BIN [18] connecting actors/actresses (nodes) based on movie appearances (edges).

Table 2: Datasets Details for Graph Classification

| Datasets | Graphs | Average nodes per graph | Average edges per graph | Classes |
|---|---|---|---|---|
| MUTAG | 188 | 17.93 | 19.79 | 2 |
| PTC | 344 | 14.29 | 14.69 | 2 |
| IMDB-BIN | 1000 | 19.77 | 193.06 | 2 |

## C.3   Baselines Details and Settings

The baseline methods are given below:

Table 3: The hyperparameters of the baselines on node classification task.

| Model | Training Type | Implementation |
|---|---|---|
| GCN [6] | Supervised | |
| GAT [15] | Supervised | https://github.com/rusty1s/pytorch_geometric |
| APPNP [7] | Supervised | |
| MagNet [21] | Supervised | https://github.com/matthew-hirn/magnet |
| DiGCN [14] | Supervised | https://github.com/flyingtango/DiGCN |
| DGI [16] | Self-supervised | https://github.com/PetarV-/DGI |
| GMI [11] | Self-supervised | https://github.com/zpeng27/GMI |
| MVGRL [4] | Self-supervised | https://github.com/kavehhassani/mvgrl |
| GraphCL [20] | Self-supervised | https://github.com/Shen-Lab/GraphCL |
| GRACE [22] | Self-supervised | https://github.com/CRIPAC-DIG/GRACE |
| GCA [23] | Self-supervised | https://github.com/CRIPAC-DIG/GCA |

For all baseline models, we use their model structure in the original papers, including layer number, activation function selection, normalization and regularization selection, etc. We implement GCN, GAT, and APPNP using PyG [2]. Note that for DiGCN, we do not use its inception module but only use the directed graph convolution. Detailed hyper-parameter settings are shown in Table 4.

To ensure the generality of the model, we have minimized the variation of hyperparameters. Our implementation is based on the GRACE code, with improvements to the topological data augmentation and the model training scheme. For the feature-level perturbation part, we also apply the dropping feature method used in GCA, GRACE and MVGRL, with the same parameters as in GRACE. We initialize our model with Glorot initialization [3] and use Adam optimizer [5] in all datasets. The

Table 4: The hyperparameters of baselines for node classification task.

| Model | layer | lr | weight-decay | hidden dimension | Others |
|---|---|---|---|---|---|
| GCN | 2 | 0.01 | 5e-4 | 64 | - |
| GAT | 2 | 0.005 | 5e-4 | CORA-ML & CITESEER:8 others:32 | heads=16 |
| APPNP | 2 | 0.01 | 5e-4 | 64 | $\alpha = 0.1$ |
| MagNet | 2 | 5e-3 | 5e-4 | 64 | $K = 1$, $q = 0.1$ |
| DiGCN | 2 | 0.01 | 5e-4 | 64 | $\alpha = 0.1$ |
| DGI | 1 | 0.001 | 0 | 512 | max-LR-iter=150 |
| GMI | 1 | 0.001 | 0 | 512 | $\alpha = 0.8$, $\beta = 1$, $\gamma = 1$ |
| MVGRL | 1 | 0.001 | 0 | 512 | $\alpha = 0.2$, $t = 5$ |
| GraphCL | 1 | 0.001 | 0 | 512 | drop rate=0.2 |
| GRACE | 2 | 0.001 | 1e-5 | CORA-ML & CITESEER:128 others:256 | augmentation parameters are consistent with the paper |
| GCA | 2 | 0.001 | 1e-5 | CORA-ML & CITESEER:128 others:256 | augmentation parameters are consistent with the paper |

initial learning rate is set to 0.001 and the weight decay factor is set to 1e-5 on all datasets. We set the number of layers used in the GCN encoder as 2. As stated in Section 3.2 of the main text, we fixed the initial and ending difficulty as 0.8 and 0 to obtain the complete contrastive information. The detailed parameter settings are shown in Table 5.

Table 5: The hyperparameters of our models.

| Our models | layers | lr | weight-decay | hidden dimension | init $\Delta\alpha$ | end $\Delta\alpha$ | epochs |
|---|---|---|---|---|---|---|---|
| CORA-ML | 2 | 0.001 | 1e-5 | 128 | 0.8 | 0 | 600 |
| CITESEER | 2 | 0.001 | 1e-5 | 128 | 0.8 | 0 | 300 |
| AM-PHOTO | 2 | 0.001 | 1e-5 | 512 | 0.8 | 0 | 2000 |
| PUBMED | 2 | 0.001 | 1e-5 | 256 | 0.8 | 0 | 600 |
| DBLP | 2 | 0.001 | 1e-5 | 256 | 0.8 | 0 | 600 |

For the graph classification task in the main text, we follow the setting in the [4] and only change the data augmentation and the pacing function. The hyperparameters are as follow.

Table 6: The hyperparameters on graph classification task.

| Method | Hyperparameters | MUTAG | PTC | IMDB-BIN |
|---|---|---|---|---|
| MVGRL | layer | 4 | 4 | 2 |
|  | batches | 256 | 128 | 256 |
|  | epochs | 20 | 20 | 20 |
|  | $\alpha = 0.2$ | 0.1 | 0.1 | 0.1 |
| MVGRL+DiGCN | layer | 4 | 4 | 2 |
|  | batches | 256 | 128 | 256 |
|  | epochs | 20 | 20 | 20 |
|  | $\Delta\alpha$ | $0.8 \rightarrow 0$ | $0.8 \rightarrow 0$ | $0.8 \rightarrow 0$ |

## C.4 Pseudocode

Here, we provide the pseudocode for two key algorithms, one for Laplacian perturbation proposed in Section 2 of the main text and the second for directed graph contrastive learning (DiGCL) introduced in Section 3 of the main text.

---

**Algorithm 1:** Laplacian Perturbation $\Phi(\cdot)$ Procedure

---

**Input:** Directed graph adjacency matrix: $\mathbf{A}$, teleport probability $\alpha$, perturbation term $\Delta\alpha$

**Output:** Perturbed Laplacian $\hat{\mathbf{L}}_{\mathrm{appr}}$

1   $\tilde{\mathbf{A}} \leftarrow \mathbf{A} + \mathbf{I}^{n \times n}$ ;

2   $\tilde{\mathbf{P}} \leftarrow \tilde{\mathbf{D}}^{-1}\tilde{\mathbf{A}}$ ;

3   $\hat{\alpha} = \alpha + \Delta\alpha$ ;

4   $\hat{\pi}_{\mathrm{appr}} \leftarrow (1-\hat{\alpha})\hat{\pi}_{\mathrm{appr}}\tilde{\mathbf{P}} + \frac{1}{n}\frac{\hat{\alpha}}{1+\hat{\alpha}}\mathbf{1}^{1 \times n}$;

5   $\hat{\mathbf{\Pi}}_{\mathrm{appr}} \leftarrow \frac{1}{\|\hat{\pi}_{\mathrm{appr}}\|_1}\mathrm{Diag}(\hat{\pi}_{\mathrm{appr}})$;

6   $\hat{\mathbf{L}}_{\mathrm{appr}} \leftarrow \mathbf{I} - \frac{1}{2}\left(\hat{\mathbf{\Pi}}_{\mathrm{appr}}^{\frac{1}{2}}\tilde{\mathbf{P}}\hat{\mathbf{\Pi}}_{\mathrm{appr}}^{-\frac{1}{2}} + \hat{\mathbf{\Pi}}_{\mathrm{appr}}^{-\frac{1}{2}}\tilde{\mathbf{P}}^{T}\hat{\mathbf{\Pi}}_{\mathrm{appr}}^{\frac{1}{2}}\right)$;

7   **return** $\hat{\mathbf{L}}_{\mathrm{appr}}$

---

 

---

**Algorithm 2:** DiGCL Training Procedure

---

**Input:** Directed graph: $\mathcal{G}$, teleport probability $\alpha$, scoring function: $\mathcal{D}$, pacing function: $\mathcal{P}$, encoder: $f^*(\cdot)$, projection head: $g(\cdot)$, data augmentation function: $\Phi(\cdot)$, loss function: $\ell(\cdot)$, number of iterations: $L$, initial difficulty $d_a$, ending difficulty $d_b$

**Output:** Trained Encoder $f^*(\cdot)$

1   **Initialize** $f^*(\cdot)$, $g(\cdot)$;

2   **for** $l \leftarrow 0$ to $L$ **do**

3      $d_m = \mathcal{P}_{(d_a,d_b)}(l)$ ;

4      $\Delta\alpha \leftarrow \mathcal{D}^{-1}(d_m)$ ;

5      $U \leftarrow \mathbf{L}_{\mathrm{appr}}(\mathcal{G}, \alpha)$ ;

6      $V \leftarrow \Phi_{\Delta\alpha}(\mathcal{G}, \alpha)$ ;

7      $\mathbf{H}_U \leftarrow f(U)$ ;

8      $\mathbf{H}_V \leftarrow f(V)$ ;

9      $\mathbf{Z}_U \leftarrow g(\mathbf{H}_U)$ ;

10      $\mathbf{Z}_V \leftarrow g(\mathbf{H}_V)$ ;

11      $\mathrm{loss} \leftarrow \ell(\mathbf{Z}_U, \mathbf{Z}_V)$ ;

12      $\mathrm{SGD(loss)}$ ;

13   **end**

14   **return** $f^*(\cdot)$

---