# OpenReview forum: "Directed Graph Contrastive Learning"
_NeurIPS.cc/2021/Conference — NeurIPS 2021 Poster_

### Official Review · Reviewer_fews · 2021-07-09

**Rating:** 5
**Confidence:** 3

**Summary:**

This paper focuses on learning with directed graphs (digraphs), and aims to address the problems with existing GCL methods that 1) the augmentations undesirably alter meaningful structural information, and 2) the number of contrastive views is limited.

As the first contribution, the authors propose a solution called "Laplacian perturbation", where contrastive views are produced through changing the approximate eigenvector by tuning the teleport probability (of page rank). Laplacian perturbation preserves structural information, can be speeded up by power method (please see my question about this in the main review), and allows to produce an infinite number of contrastive views (please see my comments in the main review).

The second contribution of this work is the DiGCL framework, which performs graph contrastive learning between views by minimizing the InfoNCE loss, a lower bound to the mutual information. The authors also use curriculum learning to help with training, where they gradually "anneal" the difficulty of the contrastive tasks.

The effectiveness of DiGCL is demonstrated on various node-classification datasets, where DiGCL performs particularly well on directed graphs as desired.

**Limitations And Societal Impact:**

As the authors explained, the main limitation is that since DiGCL is designed at node-level, its performance for graph-level tasks may not be as strong.

There is no direct societal impact for this work.

**Main Review:**

**Strength**: The proposed Laplacian perturbation addresses the problem of altering informative structural information with previous augmentation methods, and has shown superior empirical performance especially on node-level tasks on directed graphs.

**Concerns**: I do not agree with the claim about providing "an infinite number of views" and would consider the proposed method as offering two views instead, since the contrastive loss is calculated over two terms. Continuous augmentation (which in this case means varying $\Delta\alpha$) shouldn't count towards the number of views; for example, for images, we consider there to be 2 views rather than infinite views, even though color jittering is a continuous augmentation (modulo the discretization of pixel values).

**Questions**:
* About power method: the last equality of eq (5) is wrong, i.e. $\mathbf{1}^{1 \times n} \pi_{\text{appr}}^t \mathbf{1}^{n\times 1} \neq \pi_{\text{appr}}^{t}$, since matrix (vector) multiplication is not commutative. This means the power method is not applicable.

* Curriculum learning:
  * how is the number of subtasks $M$ chosen? How would its choice affect the performance, and can it be tuned?
  * What's the reason for using a uniform partition on $[0, 1-\alpha)$, as opposed to e.g. randomly sample $M$ choices of $\Delta\alpha$ in $[0, 1-\alpha)$?

**Writing & Clarity**:
* The paper is clearly situated in the literature which is appreciated.
* Sec 2 currently refers heavily to a previous work and could benefit from some cleanup.
* Definition of the scoring & pacing function: the notation should be from the domain to the image, both of which are sets, but $M, L, d$ are all numbers.
* For related work, citation [12] actually states that more views don't help with learning (e.g. they compared with 3 views in the experiments), which is different from the premise of this work that graph learning needs more views. I think the discrepancy lies in what counts as a "view"; maybe changing from "multi-view" to "multiple level of augmentation strength" is more suitable and can help clarify.

=======
Post-rebuttal update: I thank the authors for their thoughtful and detailed responses. Unfortunately I think the current version doesn't provide either strong theoretical results or compelling experiments, so would not recommend an accept as of now. I encourage the authors to resubmit though, especially given that the additional results in your responses already look promising. Best of luck!

**Time Spent Reviewing:**

3

---

> ### Author Response · Authors · 2021-08-10
> **Response to several questions**
>
> Thank you for your constructive feedback and the opportunity to answer the questions
>
> ## R1: Clarification on the number of views
>
> >I do not agree with the claim about providing "an infinite number of views" and would consider the proposed method as offering two views instead since the contrastive loss is calculated over two terms.
>
> Thank you for pointing out descriptions in the paper that may have caused confusion.
>
> We have summarized a few of our points below.
>
> 1. The phrase "an infinite number of views" is intended to emphasize the high flexibility of our model. The Laplacian perturbation is able to provide as many comparison views as required, and the multi-view contrastive learning framework is also able to learn graph structure information from all generated views. This description leads to ambiguity and will be removed in subsequent versions.
> 2. Contrastive loss is calculated in each subtask. Although there are only two terms in the contrastive loss, the **objective function of our model (Eq(9))** is the expectation of all possible contrastive views. In a continuous difficulty space, i.e. where $\alpha$ takes on infinite values, the contrastive views to be optimized are also infinite (or a large number). We agree that only two views will be trained in one subtask, which is consistent with your views, but for the main task, the number of views it can handle is the sum of all subtasks.
> 3. Due to some field gaps, the visual and the graph may have different understandings of some definitions. The initial input of our model is only **one graph**. In each epoch, this graph is subjected to a data augmentation scheme with different parameters to obtain a new contrastive graph. In other words, assuming that there are 1000 epochs, 1000 different graphs will be produced. And all these graphs will be learned by the model in one complete training session. In image contrastive learning, the datasets used often contain a very large number of images. For a single image, it is more like a node in the graph.
>
> I hope that the above clarifications will help you to better understand some of the arguments in our paper and dispel misconceptions you may have about our contribution.
>
> ## R2: Mistake about power method
>
> > About power method: the last equality of eq (5) is wrong, i.e. 11×nπapprt1n×1≠πapprt, since matrix (vector) multiplication is not commutative. This means the power method is not applicable.
>
> We apologize that we did not do the proofreading carefully, and there was indeed an error in the derivation of Eq(5). After the correction, Eq(5) is
>
>
> $$
> \pi_{a}^{t+1}=(1-\alpha)\pi_{a}^t\tilde{\mathbf{P}}+\frac{a}{n}\pi_{a}^t\mathbf{1}^{n\times n}.
> $$
>
>
> We use $a$ to replace $appr$ for simplification. Using this **acceleration algorithm** can speed up the computation of Laplacian perturbation by about 100 times. A detailed Python implementation is available in the ***code of the Supplementary material***.
>
>
>
> ## R3: Questions about curriculum learning
>
> >how is the number of subtasks M chosen?  How would its choice affect the performance, and can it be tuned?
>
> >What's the reason for using a uniform partition on [0,1−α), as opposed to e.g. randomly sample M choices of Δα in [0,1−α)?
>
> These two questions are actually the same, i.e., how we chose the step size in the paper. Choosing M is actually choosing the **step size**. Meanwhile, uniform partition on $[0,1−\alpha)$ is equivalent to set **step size** = 1 and randomly sampling M choices in $[0,1−\alpha)$  is equivalent to let **step size** randomly  sample from [0,1].
>
> As stated in Line 214-215, we set the $M$ equals to $L$, i.e., $M$ = number of epochs, which means the step size is 1. There are two reasons as follows.
>
> 1. As the difficulty space is continuous, setting the step size to 1 is the most fine-grained and stable discretization of the subtasks, i.e., each epoch fulfills a subtask.
> 2. We can change the learning speed (have the same effect with step size) by changing the slope of the pacing function, reducing the number of parameters that need to be adjusted.
>
> Through pre-setting the step size, we transform the issues of **setting M** and **partitioning subtasks** to **designing the pacing function**.
>
> Several comprehensive experiments on the pacing function are presented in the *Supplementary material Section B.2*.
>
> From these experiments, we can draw a few empirical conclusions as follows.
>
> 1. Log-based pacing function performs the best of the three pacing functions, but not too far from the other two pacing functions.
> 2. The best results are obtained by setting the start and end points to be the boundary points of the Laplacian perturbation parameter space.
> 3. The order in which the views are learned is crucial, with contrastive views working best from easy to hard. (Concluded from Table 1 in the main text)
>
> For the **starting and ending difficulty scores**，in accordance with the second conclusion, we consider that it is better to take the boundary values, which are effective and do not require parameter selection. For the **type of pacing functions**, according to the first and third conclusions, the different pacing functions have an impact on the results of the model but are not as important as the learning order. We believe that any pacing functions that satisfy the order of easy to hard can be chosen.
>
>
>
> ## R4: Improvements in writing
>
> > Sec 2 currently refers heavily to a previous work and could benefit from some cleanup.
>
> We apologize that the pre-defined theoretical sections are extensive and cause difficulties in reading. We will endeavor to clean up the redundant descriptions as much as possible.
>
> > Definition of the scoring & pacing function: the notation should be from the domain to the image, both of which are sets, but M, L,d are all numbers.
>
> Indeed, we will rectify this mistake.
>
> ## R5: Clarification on multi-view descriptions
>
> > For related work, citation [12] actually states that more views don't help with learning (e.g. they compared with 3 views in the experiments), which is different from the premise of this work that graph learning needs more views.
>
> Thank you for pointing out this conflict with our argument.  MVGRL[1] explores in the experiment whether three views could help train the model better. They use three different views from three different data augmentation methods: original graph, graph diffusion using heat kernel, and graph diffusion using PageRank. The experimental results show that using multiple views does not help the model to learn downstream tasks. It is vital to clarify the differences regarding our work and MVGRL for the definition of multi-views.
>
> ```
> While MVGRL's multi-views are multiple data augmentation methods with one parameter each, the multi-views in our paper is one data augmentation method: **Laplacian perturbation** with multiple parameters.
> ```
>
>
> The reasons why we argue that learning multiple views of one data augmentation method (Laplacian perturbation) is helpful are as follows.
>
> 1. Help the model to reduce the number of parameters that need to be predefined. As stated in Line 168-171, the choice of parameters for the contrastive view is very important and current methods require the use of grid search, for example, to find parameters. Our Laplacian perturbation does not require predefined parameters, which are controlled by the pacing function.
> 2. Increasing the stability of the model. For existing models, the quality of the contrastive views determines the effectiveness of the encoder. However, we know nothing about downstream tasks and which views are good[1]. Optimizing the performance of the model on multiple contrastive views can enhance the stability of the model and reduce the variance due to the quality of the views.
> 3. Help improve the generalization of the model. InfoMin[1] designs a multi-view strategy that iteratively maximizes the mutual information in the worst views to obtain generalized performance. They argue that learning from multiple views can help the model improve generalization since we have zero knowledge about the downstream tasks. We share similar motivations with InfoMin that optimizing the performance of the model on all possible contrastive views. Detailed explanations are given in Lines 167-178 of the main text.
>
>
> >I think the discrepancy lies in what counts as a "view"; maybe changing from "multi-view" to "multiple level of augmentation strength" is more suitable and can help clarify.
>
> Indeed, we will carefully consider how to distinguish the definition of our view from other existing papers, and how it can be clearly described.
>
> ## R6: Clarification on limitations
>
> > As the authors explained, the main limitation is that since DiGCL is designed at node-level, its performance for graph-level tasks may not be as strong.
>
> We believe that there are several reasons why our model does not perform well on graph classification problems as follows.
>
> 1. the datasets are undirected graphs, and the effect of using Laplacian perturbation on them is not as obvious as in digraphs
> 2. the graph classification problem requires more graph-level contrastive information, while our model focuses on node-level
> 3. our scoring function designed for node-level curriculum learning is not suitable for graph-level difficulty measure
>
> Based on these, we argue that it is reasonable for us to focus on node-level contrastive learning.
>
>
>
> **Thanks for your careful review!**
>
>
>
> [1] Hassani, K., & Khasahmadi, A. H. (2020, November). Contrastive multi-view representation learning on graphs. In *International Conference on Machine Learning* (pp. 4116-4126). PMLR.
>
> [2] Tian, Y., Sun, C., Poole, B., Krishnan, D., Schmid, C., & Isola, P. (2020). What makes for good views for contrastive learning?. *arXiv preprint arXiv:2005.10243*.

---

> > ### Comment · Reviewer_fews · 2021-08-24
> > **Thank you for your response**
> >
> > I appreciate the very detailed responses from the authors, which have addressed several concerns of mine.
> > I acknowledge the contributions of this work, in particular the emphasis on preserving the graph structure and the use of curriculum learning, and I agree that focusing on node-level tasks is reasonable.
> >
> > I'd like to clarify further about the number of views: the authors replied that "_Due to some field gaps, the visual and the graph may have different understandings of some definitions. The initial input of our model is only one graph._"
> >
> >  -- I agree with this statement, though I think there is not field gap and that a vision dataset can be considered the exact same way: at each epoch, each image in the dataset is applied a random augmentation (e.g. this is the practice in SimCLR), hence one can consider the vision dataset as one graph (which is different at each epoch due to augmentation), as the authors also pointed out (that each image is a node).
> >
> > This means the SimCLR-style augmentation (which is generally considered as creating _two_ views rather infinite views) and the proposed Laplacian perturbation is conceptually very similar. I hence recommend rephrasing the contribution about generating an infinite number of views, otherwise it would make the novelty of the proposed augmentation method sounds more significant than it is and can be misleading.

---

> > > ### Author Response · Authors · 2021-08-24
> > > **Thanks for feedback**
> > >
> > > Thank you very much for your response, and we are delighted that our contributions were acknowledged.
> > >
> > > We understand the vagueness and misleading nature of some of our definitions in the paper thanks to your response, e.g., generating an infinite number of views. We are deeply sorry that these were reduced to demerits.
> > >
> > > We will gratefully accept your suggestion and make certain that rebuttal content is included in the updated version based on your recommendations.
> > >
> > > **Thanks for your review!**

---

> > > ### Comment · Reviewer_WA6U · 2021-09-01
> > > **Agree with the SimCLR-style augmentation**
> > >
> > > After reading the reviews from Reviewer fews, I agreed that each epoch has a different and continuous augmented data, which is very similar to the Laplacian perturbation. I do think we call it two views make more sense to me, or at least in the CV and NLP area. Also, SimCSE uses two dropouts to create two different views from one sentence, which changes with the training epoch.

---

### Official Review · Reviewer_haDS · 2021-07-12

**Rating:** 6
**Confidence:** 4

**Summary:**

The paper proposes to adopt graph contrastive learning (GCL) on directed graphs by addressing two problems: (i) retaining structure information (specific for digraph) in augmentations; (ii) increasing views of contrasting in the CL framework.
To reach these goals, authors propose Laplacian perturbation and the digraph contrastive learning framework (DiGCL).
Experimental results verify the advantage of the proposed method.

**Limitations And Societal Impact:**

Not applicable.

**Main Review:**

(Originality) Studying GCL from the perspective of digraphs is novel for me.

(Quality) The paper is generally well-presented, with the idea clearly conveyed. Experiments are conducted on multiple compared methods which is appreciated, but it would be better if larger datasets can be examined (e.g. ogb-level of scale of graphs) since small benchmarks though are conventionally used, are easy to be overfitted for certain methods to be less convinced.
Besides, I have the following questions.

(i) The motivation to propose Laplacian perturbation as augmentation is to preserve structure information which is presumably crucial for digraphs. Experiments support this by stating that, methods with structure-disrupting augmentations (MVGRL, GraphCL...) underperform while methods without augmentations (DGI, GMI) are competitive. I would challenge this assumption that should be application-specific. The examined datasets in this paper focus on citation networks and social networks, which are usually noisy that the preserved structure knowledge might not be desired. Thus, perturbation invariance is introduced from previous GCL methods to make models more robust to generate good representations. From this sense, I am not fully persuaded of the groundedness of this assumption. Further, in DGI actually augmentations are applied if you refer to their negative pairs, which should be also structure-disrupting but still maintain good performance.

(ii) In Section 2.3 the connection between Laplacian perturbation and von Neumann entropy is drawn which is good, but in Section 3 the DiGCL framework then goes back to InfoNCE loss, which makes me feel dispartment of the story. Can authors clarify the connection of between Section 2.3 and Section 3?

**Time Spent Reviewing:**

2

---

> ### Author Response · Authors · 2021-08-10
> **Response to several questions**
>
> We appreciate the detailed and positive comment, which reflects the essential contributions of our work.
>
> ## R1: Explanation on the structure-disrupting assumption
>
> > The motivation to propose Laplacian perturbation as augmentation is to preserve structure information which is presumably crucial for digraphs. Experiments support this by stating that, methods with structure-disrupting augmentations (MVGRL, GraphCL...) underperform while methods without augmentations (DGI, GMI) are competitive. I would challenge this assumption that should be application-specific. The examined datasets in this paper focus on citation networks and social networks, which are usually noisy that the preserved structure knowledge might not be desired. Thus, perturbation invariance is introduced from previous GCL methods to make models more robust to generate good representations. From this sense, I am not fully persuaded of the groundedness of this assumption. Further, in DGI actually, augmentations are applied if you refer to their negative pairs, which should be also structure-disrupting but still maintain good performance.
>
> We are very glad to receive such an in-depth review.
>
> > I would challenge this assumption that should be application-specific.
>
> We fully agree that the different models behave differently for different subtasks, which we actually have zero knowledge of during unsupervised training.
>
> > Thus, perturbation invariance is introduced from previous GCL methods to make models more robust to generate good representations. From this sense, I am not fully persuaded of the groundedness of this assumption.
>
> We can also fully understand your point of view. You think that in some cases, such as noisy datasets, using a data augmentation method that would damage the graph structure may not reduce the effectiveness of the model but rather increase its robustness.
>
> We will elaborate on our thoughts, which are divided into the following main points.
>
> - First, we need to clarify a point. One of the reasons we believe that the data augmentation scheme needs to keep the graph structure intact is that the perturbed graph will act as the adjacency matrix for the GNN-based encoder. This means that if the graph structure is perturbed, it is not only the input data that is affected but also the structure of the encoder. This can have an impact on the stability of the model.
>
> - Second, structure-disrupting data augmentation methods need to work within a certain limit. Referring to InfoMin[1], there has a sweet spot between the data augmentation and the downstream task. Increasing the data augmentation strength before the sweet spot can provide more contrastive information, thus helping the model to learn the graph representation better. After this sweet spot, the model learns less efficiently as continuing to increase the contrastive strength can cause significant damage to the graph structure. However, we do not know where the sweet spot is. Designing data augmentation schemes that do not change the graph structure will be able to break this limit.
>
> - Third, keeping the graph structure unchanged will help us to theoretically analyze the impact of data augmentation schemes on the graph. Most data augmentation methods are designed empirically, such as randomly dropping edges/nodes. We cannot quantitatively measure the impact of these data augmentation methods on the graph structure. In contrast, Laplacian perturbation has very clear upper and lower bounds on its impact on the graph.
>
> - Finally, we argue that robustness to graph structural integrity varies across downstream tasks. For example, in experiments on node classification, there will be a higher tolerance for operations such as dropping edges than in link prediction experiments. It is worth noting that for digraphs, the accuracy of link prediction relies heavily on how much information you can retain about the direction of edges.
>
> To argue that preserve structure information does help to learn information about the direction of edges, we have added two experiments on link prediction, one is edge direction prediction and another is link existence prediction for digraphs. Due to time constraints, we provide part of the experimental results in the table below.
>
> | Direction Prediction | Cora-ML               | CiteSeer              | Existence prediction | Cora-ML               | CiteSeer              |
> | -------------------- | --------------------- | --------------------- | -------------------- | --------------------- | --------------------- |
> | GCN[4]               | $ 79.68 \pm 1.20$     | $ 68.34 \pm 0.96$     | GCN[4]               | $ 81.63 \pm 0.51$     | $ 77.12 \pm 0.71$     |
> | MVGRL[5]             | $ 82.79 \pm 0.90$     | $ 76.17 \pm 1.04$     | MVGRL[5]             | $82.50 \pm 1.12$      | $79.92 \pm 0.92$      |
> | GCA[6]               | $ 83.30 \pm 0.36$     | $ 78.53 \pm 0.88$     | GCA[6]               | $ 81.89 \pm 0.77$     | $80.37 \pm 0.61$      |
> | Ours+De              | $ 80.93 \pm 0.19$     | $ 75.32 \pm 0.64$     | Ours+De              | $ 79.80 \pm 0.71$     | $ 79.19 \pm 0.22$     |
> | Ours+Lp              | **$ 84.42 \pm 0.28$** | **$ 81.56 \pm 0.97$** | Ours+Lp              | **$ 82.61 \pm 0.49$** | **$ 81.03 \pm 0.88$** |
>
> ***De*** means using random removing edges and ***Lp*** means using Laplacian perturbation.
>
> To facilitate the comparison of experimental results, we have transcribed the *Table1 of the Supplementary material* here.
>
> | Methods | Data Augmentation Method   | CiteSeer             | Changes | Am-Photo             | Changes |
> | ------- | -------------------------- | -------------------- | ------- | -------------------- | ------- |
> | Ours    | Random removing edges      | $64.97 \pm 0.08$     | $-2.45$ | $88.45 \pm 0.01$     | $-0.96$ |
> | Ours    | **Laplacian perturbation** | **$67.42 \pm 0.14$** | $0$     | **$89.41 \pm 0.11$** | $0$     |
>
> Although we cannot directly compare the difference in accuracy between the different tasks, it is easy to see from these two tables that random dropping edges has a greater impact on the experiments for link prediction. The experimental results corroborate that the requirements for the graph structural integrity differ between subtasks. We will add the detailed experimental setup, more baselines, and datasets in subsequent versions for the link prediction task.
>
> Thank you for your suggestion and we hope that our motivation could be recognized, namely to provide sufficient contrastive information without harming the structure of the digraphs.
>
> We will revise our description, such as "preserve structure information gives better results", in subsequent versions to prevent possible misunderstandings afterward.
>
>
>
>
>
>
>
>
> ## R2: Improve the coherence of the paper
>
> > In Section 2.3 the connection between Laplacian perturbation and von Neumann entropy is drawn which is good, but in Section 3 the DiGCL framework then goes back to InfoNCE loss, which makes me feel dispartment of the story. Can authors clarify the connection of between Section 2.3 and Section 3?
>
> We apologize for the confusion in reading due to the order of the paper's content. Since the von Neumann entropy is intended to give Laplacian perturbation theory guarantees, it needs to appear in Section 2. This leads to a tear in the context of Section2 and 3. There is actually no story connection between Section 2.3 and Section 3.
>
> We will carefully consider the layout of the paper and improve it in subsequent versions.
>
>
>
> ## R3: Improve experiments
>
> > it would be better if larger datasets can be examined (e.g. ogb-level of scale of graphs)
>
>  We are fully aware that the results of experiments on large-scale datasets are more convincing. However, most graph contrastive learning baselines do not support large datasets[1,2,3]. MVGRL[3], for example, suffers from OOM problems on DBLP with only 100k edges. The smallest ogb dataset, ogbn-arxiv, has 10m edges. We will provide experimental results for larger datasets in a subsequent version. The dataset may not be comparable in size to ogb but will be as close as possible.
>
>
>
> Besides, we have investigated the augmentation time of Laplacian perturbation on randomly generated datasets of different graph sizes. The experimental results are presented in **Figure 3(b) in the main text**. We can find that even with 2M edges, our proposed Laplacian perturbation takes about 50ms to calculate the perturbed Laplacian matrix. This is made possible by the acceleration algorithm we propose in **Section 2.2 in the main text**. In addition to the manually generated datasets, we will include time complexity experiments on large-scale real datasets in a subsequent version.
>
>
> **Thanks for your review!**
>
>
> [1] Tian et al., What makes for good views for contrastive learning?.

---

### Official Review · Reviewer_ZF6P · 2021-07-16

**Rating:** 4
**Confidence:** 4

**Summary:**

This work presents a multi-view digraph contrastive learning framework. It can generate contrastive views by Laplacian perturbation and learn from them. The model is trained by the multi-task curriculum learning to progressively learn from multiple easy-to-hard contrastive views. Experiments on several datasets show the effectiveness.

**Limitations And Societal Impact:**

This work cannot solve the graph-level problems, such as pre-training on protein molecular, drug property prediction, etc. This work needs to manually design the pacing function to preplan the training speed, such as linear or exponential functions.


**Main Review:**

This work proposes the contrastive learning algorithm on directed graphs. To learn the characteristics of digraphs, this work first generates an arbitrary number of contrastive views with Laplacian perturbation. Then, it uses curriculum learning strategy to optimize the objective function. Through experiments, the authors empirically show the method outperforms various baselines.

The strengths of this work are as follows. 1) The research problem is important. Contrastive learning on digraphs is relatively under-explored in previous works. 2) The paper is relatively well-written and the ideas are clearly clarified. The figures illustrate the model framework clearly. 3) The proposed method achieves better results over various baselines on some datasets. The baselines considered in the experiments are extensive.

I also have some concerns/questions on this work: 1) The motivation is not very clear. Why is it insufficient to use a small number of views? Why is the multi-task curriculum learning adopted in the method? What is the primary difference between contrastive learning on undirected and directed graphs? 2) The novelty and technical contributions of this work are limited. Although the theoretical part of the method is correct to me, it is incremental and similar to related works (see Digraph Inception Convolutional Networks. NeurIPS 2020). The data augmentation strategy using Laplacian perturbation is trivial. 3) The experiments are not convincing enough. To validate the performance, the authors choose few and small graph datasets. It can be more convincing if some popular larger datasets are considered (such as Open Graph Benchmark). More analysis of the experimental results should also be present to give more insights to the readers. How much performance gain comes from the part of the model design and the curriculum learning strategy?


**Time Spent Reviewing:**

10

---

> ### Author Response · Authors · 2021-08-10
> **Response on motivation, contribution and experiments**
>
> Thank you for your invaluable feedback and the opportunity to articulate our motivation and contribution.
>
> ## R1: Motivation of our work
>
> > The motivation is not very clear.
>
> ### Why is it insufficient to use a small number of views?
>
> Firstly, we want to clarify a definition that may be misleading. The multi-views in our paper is one data augmentation method: **Laplacian perturbation** with multiple parameters.
>
> The reasons why we argue that learning multiple views of one data augmentation method (Laplacian perturbation) is helpful are as follows.
>
> 1. Help the model to reduce the number of parameters that need to be predefined. As stated in Line 168-171, the choice of parameters for the contrastive view is very important and current methods require the use of grid search, for example, to find parameters. Our Laplacian perturbation does not require predefined parameters, which are controlled by the pacing function.
> 2. Increasing the stability of the model. For existing models, the quality of the contrastive views determines the effectiveness of the encoder. However, we know nothing about downstream tasks and which views are good[1]. Optimizing the performance of the model on multiple contrastive views can enhance the stability of the model and reduce the variance due to the quality of the views.
> 3. Help improve the generalization of the model. InfoMin[1] designs a multi-view strategy that iteratively maximizes the mutual information in the worst views to obtain generalized performance. They argue that learning from multiple views can help the model improve generalization since we have zero knowledge about the downstream tasks. We share similar motivations with InfoMin that optimizing the performance of the model on all possible contrastive views. Detailed explanations are given in Lines 167-178 of the main text.
>
> ### Why is the multi-task curriculum learning adopted in the method?
>
> We are the **first** work to leverage multi-task curriculum learning to address the problem of multi-view graph contrastive learning. As illustrated in Line 182-186,  multi-task curriculum learning is designed to optimize the multi-view objective function proposed in Eq(10).
>
> To increase the generalization capability of the model, we propose a more general contrastive learning objective function in Eq(9).  However, maximizing Eq. (9) is not easy. We come up with the divide-and-conquer strategy that breaks down the multiple views in the objective function into multiple subtasks. A curriculum learning approach is then used to learn all the subtasks from easy-to-hard views.
>
> ### What is the primary difference between contrastive learning on undirected and directed graphs?
>
> We believe that the biggest difference is the ability to learn information about the direction of the edges.
>
> To argue that our method does help to learn information about the direction of edges, we have added two experiments on link prediction, one is edge direction prediction and another is link existence prediction for digraphs. Due to time constraints, we provide part of the experimental results in the table below.
>
> | Direction Prediction | Cora-ML               | CiteSeer              | Existence prediction | Cora-ML               | CiteSeer              |
> | -------------------- | --------------------- | --------------------- | -------------------- | --------------------- | --------------------- |
> | GCN[4]               | $ 79.68 \pm 1.20$     | $ 68.34 \pm 0.96$     | GCN[4]               | $ 81.63 \pm 0.51$     | $ 77.12 \pm 0.71$     |
> | MVGRL[5]             | $ 82.79 \pm 0.90$     | $ 76.17 \pm 1.04$     | MVGRL[5]             | $82.50 \pm 1.12$      | $79.92 \pm 0.92$      |
> | GCA[6]               | $ 83.30 \pm 0.36$     | $ 78.53 \pm 0.88$     | GCA[6]               | $ 81.89 \pm 0.77$     | $80.37 \pm 0.61$      |
> | Ours                 | **$ 84.42 \pm 0.28$** | **$ 81.56 \pm 0.97$** | Ours                 | **$ 82.61 \pm 0.49$** | **$ 81.03 \pm 0.88$** |
>
> It is easy to see that our model is able to learn digraph structural information, in particular the direction of the edges. We will add the detailed experimental setup, more baselines, and datasets in subsequent versions for the link prediction task.
>
> ## R2: Contribution and Novelty
>
> We apologize that we could not present our contributions separately and clearly, which causes the misunderstanding that our work contribution is incremental. We will elaborate on the difference between our model and existing methods and why our method is not just a simple incremental combination.
>
> ### Contributions compared to digraph Laplacian（Tong et al.）
>
> > It is incremental and similar to related works.
>
> Indeed, the **theoretical basis** of Laplacian perturbation is the digraph Laplacian matrix, and it has been studied in several works [1,2]. We argue that the digraph Laplacian is a **fundamental work**, just as using GCN[4] will **inevitably** make use of the Laplacian matrix. We summarize a few differences and improvements, compared to Tong et al.
>
> 1. Tong et al. is a supervised learning method for digraphs, whereas our work is a contrastive (self-supervised) learning approach, which is the biggest difference.
> 2. Our Laplacian perturbation is a **data augmentation method**, whereas Tong et al. is a **graph convolution operation**. The types and application scenarios of the two methods are different.
> 3. We have designed an acceleration algorithm in Section 2.2 which can speed up the computation of digraph Laplacian by about 100 times, compared to Tong et al.. The experimental results are presented in Figure 3(c).
> 4. In Section 2.3, we theoretically analyze the variation of the digraph Laplacian with teleport probability $\alpha$, which is not available in Tong et al..
>
> ### Contributions compared to other data augmentation methods
>
> > The data augmentation strategy using Laplacian perturbation is trivial.
> > Moreover, our approach has significant contributions to the **machine learning community**, compared to existing data augmentation methods.
>
> 1. Laplacian perturbation is the **first** data augmentation method designed specifically for **digraphs** to provide contrastive information **without** changing the digraph structure. It can preserve the directed structure information and minimize the impact on the subsequent GNN-based encoder.
> 2. Laplacian perturbation has **theoretical guarantees**, while most other data augmentation methods are designed empirically, such as randomly dropping edges/nodes. We cannot quantitatively measure the impact of these data augmentation methods on the graph structure. In contrast, Laplacian perturbation has very clear upper and lower bounds on its impact to the graph.
>
> Based on all the above comparisons with Tong et al. and other data augmentation methods, our contributions and improvements are significant, which may be inappropriate to treat as an increment or trivial work.
>
> ### R3: Experiments
>
> > The experiments are not convincing enough.
>
> ### Experiments on large-scale datasets
>
> Your concern is indeed one that we have considered as well, regarding the performance of Laplacian perturbation with large-scale datasets. We investigate the augmentation time of Laplacian perturbation on randomly generated datasets of different graph sizes. The experimental results are presented in **Figure 3(b) in the main text**. We can find that even with 2M edges, our proposed Laplacian perturbation takes about 50ms to calculate the perturbed Laplacian matrix. In addition to the manually generated datasets, we will include time complexity experiments on large-scale real datasets in a subsequent version.
>
> Besides, we are fully aware that the results of experiments on large-scale datasets are more convincing. However, most graph contrastive learning baselines do not support large datasets[1,2,3]. MVGRL[3], for example, suffers from OOM problems on DBLP with only 100k edges. The smallest ogb dataset, ogbn-arxiv, has 10M edges. We will provide experimental results for larger datasets in a subsequent version.
>
> ### How much performance gain comes from the part of the model design and the curriculum learning strategy?
>
> Our model can be divided into two parts, the data augmentation method: **Laplacian perturbation**, and the learning framework: **DiGCL**. We will elaborate on the performance of each part.
>
> We have summarized the results of ***Table1 of the main text*** and  ***Table1 of the Supplementary material*** as follows.
>
> | Methods             | Data Augmentation Method   | CiteSeer             | Changes | Am-Photo             | Changes |
> | ------------------- | -------------------------- | -------------------- | ------- | -------------------- | ------- |
> | Ours + No Curr + Lp | **Laplacian perturbation** | $66.99 \pm 0.54$     | $-0.43$ | $87.32 \pm 0.14$     | $-2.09$ |
> | Ours + Curr + No Lp | Random removing edges      | $64.97 \pm 0.08$     | $-2.45$ | $88.45 \pm 0.01$     | $-0.96$ |
> | Ours + Curr + Lp    | **Laplacian perturbation** | **$67.42 \pm 0.14$** | $0$     | **$89.41 \pm 0.11$** | $0$     |
>
> ***Curr*** means learning views from easy-to-hard, ***Lp*** means using Laplacian perturbation. The experimental results show that using Laplacian perturbation can help our model improve its performance in digraphs, comparing with random removing edges. Moreover, the effect can be further improved by combining Laplacian perturbation and curriculum learning.
>
> **Thanks for your review!**
>
> [1] Tian et al., What makes for good views for contrastive learning?.
>
> [2] Tong et al., Digraph inception convolutional networks.
>
> [3] Chung et al., Laplacians and the Cheeger inequality for directed graphs.
>
> [4] Kipf et al., Semi-supervised classification with graph convolutional networks.
>
> [5] Hassani et al., Contrastive multi-view representation learning on graphs.
>
> [6] Zhu et al., Graph contrastive learning with adaptive augmentation.
>
> [7] Tian et al., Contrastive multiview coding.

---

> > ### Comment · Reviewer_ZF6P · 2021-08-30
> > **Thank you for the rebuttal**
> >
> > Thank you for the responses.
> >
> > The response R1 addresses my concern on the motivation of this work, and my concerns are partially addressed by R2 and R3.
> >
> > The responses to my questions are detailed, but two important concerns are yet not well addressed to me.
> >
> > 1) The novelty and contributions are not significant. As clarified in the responses, the biggest difference between [1] and this work is that [1] is a supervised method and the proposed method is a self-supervised method. However, adopting similar methodology and changing the training manner from supervised learning to self-supervised learning are somewhat trivial to me. Although the acceleration algorithm for speeding up the computation sounds good to me, the main contributions are still not impressive.
> >
> > 2) The experiments are not convincing enough. The large-scale datasets should be considered for comparison. The authors also noticed the experiments on large-scale datasets are important, but there are no empirical evaluations in the submission. It is unclear to me why most graph contrastive learning baselines do not support large datasets as claimed in the paper. The authors should provide the explanations from theoretical or empirical views. MVGRL indeed suffers from OOM problem, how about other baselines with appropriate hyper-parameters?
> >
> > [1] Digraph Inception Convolutional Networks. NeurIPS 2020

---

> > > ### Author Response · Authors · 2021-08-31
> > > **Response to the feedback**
> > >
> > > Thank you very much for your response, and we are delighted that our contributions were recognized, which is very encouraging.
> > >
> > > > **The responses to my questions are detailed, but two important concerns are yet not well addressed to me.**
> > >
> > > We would like to provide further clarification on the above two issues and hope to allay your concerns.
> > >
> > > ## Concern about the incremental part with digraph Laplacian（Tong et al.）
> > >
> > > > However, adopting a similar methodology and changing the training manner from supervised learning to self-supervised learning is somewhat trivial to me.
> > >
> > > We fully understand your concerns, and that is why we want to re-emphasize the following points with you.
> > >
> > > 1. We did not **only** change the supervision method compared with Tong et al. [1] and treat this as a contribution. The difference in training methods is dictated by the task, and how to make good use of unlabeled data is a problem to be explored by self-supervised (contrastive) methods.
> > > 2. Tong et al. is the **theoretical basis for spectral analysis of digraph convolution**. We cannot avoid using the definition of Tong et al. when we use the digraph Laplacian in our paper. This operation is similar to the use of GCN in **MVGRL[2], GraphCL[3], GRACE[4], and GCA[5]**, but this does not prevent them from contributing to the graph contrastive learning community.
> > > 3. The proposed **Laplacian perturbation**  is the **first** data augmentation method designed specifically for digraphs and the **first** data augmentation method generating contrastive views without changing the structure of the digraph/graph. Meanwhile, it has **theoretical** guarantee.
> > >
> > > A great deal of thoughtful design and refinement has been done for contrastive learning based on Tong et al.  We wish you will support our view that this is not a trivial improvement.
> > >
> > >
> > >
> > > ## Concern about the experiments in large-scale graphs
> > >
> > > We will elaborate our views in the following three parts to dispel your concerns.
> > >
> > > ### Additional experiments under medium-sized dataset
> > >
> > > > The experiments are not convincing enough. The large-scale datasets should be considered for comparison. The authors also noticed the experiments on large-scale datasets are important, but there are no empirical evaluations in the submission.
> > >
> > > We appreciate your suggestions. **Due to hardware limitations**, we can only test our model on a medium-sized **Amazon-Computer (with 287,209 edges)** dataset in order to demonstrate that our model can be competitive on larger datasets.
> > >
> > > | Dataset         | GCN               | DiGCN[1]          | GRACE[4]          | GCA[5]            | Ours                  |
> > > | --------------- | ----------------- | ----------------- | ----------------- | ----------------- | --------------------- |
> > > | Amazon-Computer | $ 79.43 \pm 1.03$ | $ 84.98 \pm 0.27$ | $ 83.62 \pm 0.16$ | $ 85.33 \pm 0.18$ | **$ 86.50 \pm 0.31$** |
> > >
> > > We apologize that we cannot find a GPU with larger memory in a short period of time. We will add the analysis of our model performance under large-scale datasets in a subsequent version.
> > >
> > > ### Analysis of the reasons why current methods fail to handle large-scale graph
> > >
> > > > It is unclear to me why most graph contrastive learning baselines do not support large datasets as claimed in the paper. The authors should provide explanations from theoretical or empirical views.
> > >
> > > Let us first clarify one point: **We focus on the contrastive learning of directed graphs, and investigate the GCL frameworks for the large-scale graphs is not the focus of our paper**.
> > >
> > > As we addressed in the previous rebuttal, we think that there are several reasons why current graph contrastive learning baselines have difficulty processing large datasets.
> > >
> > > 1. Due to the need to compare the readout of different views,  multiple GNN-based encoders will be trained simultaneously in the model. Such as MVGRL[2] have 4 encoders, GRACE[4] and GCA[5] have 2 encoders. Multiple encoders can cause a significant increase in memory consumption.
> > >
> > > 2. The data augmentation approach may consume a lot of computational resources. For example, MVGRL[2] uses graph diffusion methods to generate contrastive views, where the PageRank kernel-based approach requires the computation of an inverse matrix. With large-scale graph datasets, this requires significant computational resources.
> > >
> > > 3. We need to calculate the distance between each node and their positive and negative samples to compute the loss. In the case of a large number of nodes, calculating the loss requires a large amount of memory consumption.
> > >
> > > ### Additional experiments of baselines' running time with different graph size
> > >
> > > > MVGRL indeed suffers from OOM problem, how about other baselines with appropriate hyper-parameters?
> > >
> > > As requested, we test the contrastive learning baselines on the generative dataset for running time on the large-scale graph. To generate arbitrary size, we construct a simple random graph with N nodes and assign 10N directed edges uniformly at random. Due to time constraints, we compared with **GRACE[4]** and **GCA[5]**, the two models with the accuracy similar to ours. We provide the training time per epoch (ms) in the table below.
> > >
> > > | Number of Nodes | 5K    | 10K    | 15K    | 20k    | 50K  |
> > > | --------------- | ----- | ------ | ------ | ------ | ---- |
> > > | GRACE[4]        | 67.35 | 140.55 | 222.92 | 342.98 | OOM  |
> > > | GCA[5]          | 71.97 | 143.34 | 257.40 | 361.34 | OOM  |
> > > | Ours            | 73.05 | 153.29 | 261.41 | 390.68 | OOM  |
> > >
> > > **\* The above baselines are implemented using the official Github repos, and the specific code links are in *Table 4 of Supplementary material*. We use 12GB TitanV GPU.**
> > >
> > > We can find that processing 20,000 nodes and 200,000 edges with 12 GB memory is the limit for these three models. Since our model requires a Laplacian perturbation operation, this will take more time than the data augmentation performed by GCA and GRACE.
> > >
> > >
> > > ### Summary
> > >
> > > As an important research direction for the graph learning community, how to optimize the performance of models on large-scale datasets has been a goal pursued by researchers. We realize that our model, like many graph contrastive learning models, has limitations in handling large-scale graph data. We will add support for large-scale datasets, such as ogb in the subsequent version, by adding hardware, replacing the encoder with one more suitable for large-scale datasets[6], or adopting a new graph learning library[7].
> > >
> > > Besides, we will ensure that rebuttal content is included in the updated version based on your suggestions.
> > >
> > > **Since we have solved most of your concerns in our previous rebuttal and you have acknowledged the contributions, we hope you will be in favor of our work after reading this clarification.**
> > >
> > > **Thanks for your review!**
> > >
> > >
> > >
> > > [1] Tong, Z., Liang, Y., Sun, C., Li, X., Rosenblum, D., & Lim, A. (2020). Digraph inception convolutional networks. *Advances in neural information processing systems*, *33*.
> > >
> > > [2] Hassani, K., & Khasahmadi, A. H. (2020, November). Contrastive multi-view representation learning on graphs. In *International Conference on Machine Learning* (pp. 4116-4126). PMLR.
> > >
> > > [3] You, Y., Chen, T., Sui, Y., Chen, T., Wang, Z., & Shen, Y. (2020). Graph contrastive learning with augmentations. *Advances in Neural Information Processing Systems*, *33*, 5812-5823.
> > >
> > > [4] Zhu, Y., Xu, Y., Yu, F., Liu, Q., Wu, S., & Wang, L. (2020). Deep graph contrastive representation learning. *arXiv preprint arXiv:2006.04131*.
> > >
> > > [5] Zhu, Y., Xu, Y., Yu, F., Liu, Q., Wu, S., & Wang, L. (2021, April). Graph contrastive learning with adaptive augmentation. In *Proceedings of the Web Conference 2021* (pp. 2069-2080).
> > >
> > > [6] Zeng, H., Zhou, H., Srivastava, A., Kannan, R., & Prasanna, V. (2019). Graphsaint: Graph sampling based inductive learning method. *arXiv preprint arXiv:1907.04931*.
> > >
> > > [7] Fey, M., Lenssen, J. E., Weichert, F., & Leskovec, J. (2021). GNNAutoScale: Scalable and Expressive Graph Neural Networks via Historical Embeddings. *arXiv preprint arXiv:2106.05609*.

---

### Official Review · Reviewer_WA6U · 2021-07-17

**Rating:** 5
**Confidence:** 3

**Summary:**

The paper describes a self-supervised framework called Laplacian perturbation to generate multiview, which can preserve the intrinsic graph structure. Unlike the original contrastive learning, the multi-view digraph contrastive learning framework can learn from all possible contrastive views generated by Laplacian perturbation. Training the contrastive learning in a multi-task curriculum learning way can progressively learn from multiple easy-to-hard contrastive views.

**Limitations And Societal Impact:**

The authors have adequately addressed the limitations.

**Main Review:**

Positives:
- It is novel to train contrastive learning progressively in a curriculum learning way, and the experiments validate that this is a very effective modification.

- Using perturbation to create data augmentation is an interesting insight. I feel it is somehow related to SimCSE, applying dropout to create noises. Also, the progressively changing levels of perturbation automatically push the model to learn from easy to difficult contrastive views is also new to me. It could be applied in other contrastive learning like to increase the dropout rate to a larger proportion.

Concerns:
- The caption for Figure 1 is not illustrative enough for understanding the overall structure. For example, how do U and V come from the pacing function? Why one node is red-colored and the other ones are blue-colored? I understand that every component is written in subsections clearly, but this figure should serve as the most important and informative part of the work in this paper.

- Equation 10, the expectation is taken over \Delta \alpha1 and \Delta\alpha2, then maximize the whole objective function over f, right? It is not equivalent to what it said below: it requires the obtained optimal encoder works well with \forall \Delta \alpha1 and \Delta\alpha2 \in [0, 1 -\alpha) to learn a more balanced and generalized representation. For all alpha1 and alpha2 should be equivalent to taking maximum over all \Delta \alpha1 and \Delta\alpha2 \in [0, 1 -\alpha). Then they take one of the views unperturbed, which deviates a little bit from the original objective, I feel it is possible to derive some theoretical guarantees in this change.

- Laplacian perturbation part involves too many mathematical formulations, which is not easy to follow. Maybe the authors could provide some examples to illustrate the change before and after Laplacian perturbation, and move some of the theoretical justification parts to supplementary.

- I feel that easy subtasks and hard subtasks are not a pair. Most of the papers I have read would say easy subtasks and difficult subtasks, or something like soft negatives and hard negatives.

- I am not very familiar with graph learning, the results this paper present is like the performance of unsupervised learning can surpass those supervised learning, which is not common in the vision area. Can you elaborate more on this phenomenon? And I found out that in some other papers, like GRACE, the results on CITESEER is 72.1±0.5, which is much higher than 61.20 ± 0.20 reported here. Maybe it is due to the different network architecture, I feel some clarification should be made to explain the performance gap.

**Time Spent Reviewing:**

6hours

---

> ### Author Response · Authors · 2021-08-10
> **Response to several concerns**
>
> We appreciate the detailed and positive comment, which reflects the essential contributions of our work.
>
> ## R1: Improvements for Figure1
>
> > The caption for Figure 1 is not illustrative enough for understanding the overall structure. I understand that every component is written in subsections clearly, but this figure should serve as the most important and informative part of the work in this paper.
>
> We apologize for the unclear design of the framework figure and the ambiguity of the caption.
>
> > For example, how do U and V come from the pacing function?
>
> After the scoring function, the pair of U and V will have a difficult score. And then,  U and V will be input into the model from easy to difficult, which is controlled by the pacing function.
>
> > Why one node is red-colored and the other ones are blue-colored?
>
> Sorry for the misunderstanding, the different color has no special meaning and we will change it to the same.
>
> We will improve the figure and its caption in subsequent versions.
>
>
>
> ## R2: Concern about simplification in Eq(10)
>
> >Equation 10, the expectation is taken over \Delta \alpha1 and \Delta\alpha2, then maximize the whole objective function over f, right? It is not equivalent to what it said below: it requires the obtained optimal encoder works well with \forall \Delta \alpha1 and \Delta\alpha2 \in [0, 1 -\alpha) to learn a more balanced and generalized representation. For all alpha1 and alpha2 should be equivalent to taking maximum over all \Delta \alpha1 and \Delta\alpha2 \in [0, 1 -\alpha). Then they take one of the views unperturbed, which deviates a little bit from the original objective, I feel it is possible to derive some theoretical guarantees in this change.
>
> We fully agree with the reviewer's concern that taking one of the views unperturbed in Eq(10) may make some difference from the original objective.
>
> There are currently two options for designing a graph contrastive learning framework, one is represented by GraphCL[1], GRACE[2], and GCA[3], where models learn representation from varying contrastive views (no original graph).  There is also another one, represented by MVGRL[4] and DGI[5], where models learn from contrastive views and the original graph. Both routes are able to achieve good performance in graph contrastive learning.
>
> We consider fixing a view in Eq. (10) as a switch from the first to the second route. As we have explained in Line 179-182, if both perturbation terms are changed, this will have an impact on the stability of the model.  We will include an explanation of this simplification, both experimentally and theoretically, in a subsequent version.
>
>
>
> ## R3: Provides examples for Laplacian perturbation
>
> > Laplacian perturbation part involves too many mathematical formulations, which is not easy to follow. Maybe the authors could provide some examples to illustrate the change before and after     Laplacian perturbation, and move some of the theoretical justification parts to supplementary.
>
> Thank you very much for your suggestion and we are aware that the part about Laplacian perturbations is abstract and giving examples would make it better understood. We will modify this in a subsequent version and include some simple and intuitive examples.
>
>
>
> ## R4: Naming issues
>
> > I feel that easy subtasks and hard subtasks are not a pair. Most of the papers I have read would say easy subtasks and difficult subtasks, or something like soft negatives and hard negatives.
>
> Indeed, easy-to-difficult and easy-to-hard are the same things in this paper. Due to space constraints, we find that using 'difficult' made typography difficult in some places, which is why we used the synonym 'hard' as an alternative. We will modify this in a subsequent version.
>
>
>
> ## R5: Explanation about the performance gap
>
> > I am not very familiar with graph learning, the results this paper present is like the performance of unsupervised learning can surpass those supervised learning, which is not common in the vision area. Can you elaborate more on this phenomenon?
>
> As we state in Line 282-284, the setup of our experiments is the same as DGI[5], MVGRL[4], GCA[3], etc. First, we train the graph encoder by contrastive learning (unsupervised) and then using the node feature generated by the graph encoder for $\ell_2$-regularized logistic regression prediction, which is supervised. This is why unsupervised graph representations can perform as well or better than supervised ones. By using a simple supervised add-on, we can measure how well a graph encoder learned through contrastive learning.
>
> > And I found out that in some other papers, like GRACE, the results on CITESEER is 72.1±0.5, which is much higher than 61.20 ± 0.20 reported here. Maybe it is due to the different network architecture, I feel some clarification should be made to explain the performance gap.
>
> The division of the dataset causes this performance gap. In the GRACE[2] they divided the dataset into the test, val, train: 70%, 20%, 10%. In our experiments, we refer to the GCN's[6] data division for the semi-supervised node classification task. The label rate is only about 0.5%-5%. GRACE is able to reach more labeled nodes during training, resulting in high prediction accuracy.
>
> We detail the task definition, dataset, and experimental setting in **Section C of Supplementary material**.
>
>
>
> **Thanks for your careful review!**
>
>
>
> [1] You, Y., Chen, T., Sui, Y., Chen, T., Wang, Z., & Shen, Y. (2020). Graph contrastive learning with augmentations. *Advances in Neural Information Processing Systems*, *33*, 5812-5823.
>
> [2] Zhu, Y., Xu, Y., Yu, F., Liu, Q., Wu, S., & Wang, L. (2020). Deep graph contrastive representation learning. *arXiv preprint arXiv:2006.04131*.
>
> [3] Zhu, Y., Xu, Y., Yu, F., Liu, Q., Wu, S., & Wang, L. (2021, April). Graph contrastive learning with adaptive augmentation. In *Proceedings of the Web Conference 2021* (pp. 2069-2080).
>
> [4] Hassani, K., & Khasahmadi, A. H. (2020, November). Contrastive multi-view representation learning on graphs. In *International Conference on Machine Learning* (pp. 4116-4126). PMLR.
>
> [5] Veličković, P., Fedus, W., Hamilton, W. L., Liò, P., Bengio, Y., & Hjelm, R. D. (2018). Deep graph infomax. *arXiv preprint arXiv:1809.10341*.
>
> [6] Kipf, T. N., & Welling, M. (2016). Semi-supervised classification with graph convolutional networks. *arXiv preprint arXiv:1609.02907*.

---

> ### Comment · Reviewer_WA6U · 2021-09-01
> **Update on the review**
>
> Thanks for the clarification. The response R2 addresses my concern.
> After reading all the reviews, I do find the infinite views is misleading, which is actually a two-view setting under CV and NLP setting. According to the ablation study, curriculum learning delivers a marginal improvement that might be caused by some randomness. Although the Laplacian perturbation is interesting, based on the not convincing experiment result mentioned by reviewer ZF6P, I found the novelty of this paper is limited.

---

> > ### Author Response · Authors · 2021-09-01
> > **Response to the updated feedback**
> >
> > > Thanks for the clarification. The response R2 addresses my concern.
> >
> > We are glad to know that our responses have helped you understand the paper.
> >
> > ## Difference with two view augmentation
> >
> > > After reading all the reviews, I do find the infinite views is misleading, which is actually a two-view setting under CV and NLP setting.
> >
> > Thank you for your suggestions. We realize that the difference in definition from CV, NLP may lead to some misinformation.
> >
> > In addition to that, we would like to clarify one point to you. **We do not claim that generating continuous contrastive views is a contribution of our method Laplacian perturbation.**  **We believe that it is how the generated continuous view is used that makes the most important contribution.** For this part we design the multi-task curriculum learning method.
> >
> > As illustrated in **Line 182-186 of the main text**,  multi-task curriculum learning is designed to optimize the multi-view objective function proposed in Eq(10). We design a short story between the different parts of the model to make it easier for the reviewer to understand.
> >
> > ~~~gfm
> > Laplacian perturbation->DiGCL: Hello framework, I am able to generate a lot of contrastive views, how do you learn from them?
> > DiGCL -> Laplacian perturbation: Using multi-view objective(Eq(9))
> > Multi-view objective->DiGCL: I am good at it, but I cannot optimize so many views.
> > DiGCL->Multi-view objective: Divide and conquer, split into subtasks and solve them one by one.
> > Multi-view objective->DiGCL: Good, but which subtask should I solve first?
> > DiGCL->Multi-view objective: Do you know the difficulty of the different subtasks?
> > Multi-view objective->DiGCL: Yes, Laplacian perturbation has told me.
> > DiGCL->Multi-view objective: Great, then you can seek help from multi-task curriculum learning.
> > Multi-task curriculum learning -> ALL: No problem, I am able to learn all contrastive views progressively from easy to hard.
> > ~~~
> >
> > We are the **first** work to leverage multi-task curriculum learning to address the problem of multi-view graph contrastive learning.  To increase the generalization capability of the model, we propose a more general contrastive learning objective function in Eq(9).  However, maximizing Eq. (9) is not easy. We come up with the divide-and-conquer strategy that breaks down the multiple views in the objective function into multiple subtasks. A curriculum learning approach is then used to learn all the subtasks from easy-to-hard views.
> >
> > We believe that even if a subtask has 2 views, on the basis of a multi-task curriculum learning, the overall number of views will still be close to infinite as long as there are enough split subtasks.
> >
> > We will remove these misunderstandings as suggested by you and the reviewer *fews*.
> >
> >
> >
> > ## The effeteness of curriculum learning
> >
> > > According to the ablation study, curriculum learning delivers a marginal improvement that might be caused by some randomness.
> >
> > Thank you for your consideration but your comment makes us feel a little confused.
> >
> > In your [previous comments](https://openreview.net/forum?id=s6JD_xBS31&noteId=8bseQ-i3jr2), you agreed with the effectiveness of our curriculum learning method.
> >
> > ```
> > It is novel to train contrastive learning progressively in a curriculum learning way, and the experiments validate that this is a very effective modification.
> > ```
> >
> > We are not sure what made you change your mind and think that the gain of our approach is due to **randomness**?
> >
> > To allay your doubts about the performance, we want to show you some experimental results.
> >
> > We have summarized the results of ***Table1 of the main text*** and  ***Table1 of the Supplementary material***, and add another dataset. The results are as follows.
> >
> > | Methods             | Data Augmentation Method   | Cora-ML              | Changes     | CiteSeer             | Changes     | Am-Photo             | Changes     |
> > | ------------------- | -------------------------- | -------------------- | ----------- | -------------------- | ----------- | -------------------- | ----------- |
> > | Ours + No Curr + Lp | **Laplacian perturbation** | $75.86 \pm 0.09$     | **$-1.67$** | $66.99 \pm 0.54$     | **$-0.43$** | $87.32 \pm 0.14$     | **$-2.09$** |
> > | Ours + Curr + No Lp | Random removing edges      | $75.74 \pm 0.42$     | $-1.79$     | $64.97 \pm 0.08$     | $-2.45$     | $88.45 \pm 0.01$     | $-0.96$     |
> > | Ours + Curr + Lp    | **Laplacian perturbation** | **$77.53 \pm 0.14$** | $0$         | **$67.42 \pm 0.14$** | $0$         | **$89.41 \pm 0.11$** | $0$         |
> >
> > ***Curr*** means learning views from easy-to-hard, ***Lp*** means using Laplacian perturbation. The experimental results show that using Laplacian perturbation can help our model improve its performance in digraphs, comparing with random removing edges. Moreover, the effect can be further improved by combining Laplacian perturbation and curriculum learning.
> >
> > On all three datasets, the use of curriculum learning is able to improve the model's effectiveness, this **consistency** that we believe should not be classified as randomness.
> >
> > ## Clarification on experiment results
> >
> > > Although the Laplacian perturbation is interesting, based on the not convincing experiment result mentioned by reviewer ZF6P, I found the novelty of this paper is limited.
> >
> > First, to clarify a key point, **reviewer ZF6P endorses our experimental results, only that we need to supplement our experiments on large-scale graph datasets.**
> >
> > As suggested, we have listed the following experiments and analyses.
> >
> > 1. **Due to hardware limitations**, we add the performance of our model with other graph contrastive learning methods under a medium-sized dataset (**Amazon-Computer with 287,209 edges**). [Experimental results here.](https://openreview.net/forum?id=s6JD_xBS31&amp;noteId=27jNzDx695)
> >
> > 2. The running time of our Laplacian perturbation is compared with other data augmentation schemes in the case of large-scale graph data. Experimental results in Figure 3 (b) of the main text.
> >
> > 3. The running time of our method compared to other graph contrastive learning frameworks under different graph size. [Experimental results here.](https://openreview.net/forum?id=s6JD_xBS31&amp;noteId=27jNzDx695)
> >
> > 4. Analysis of the reasons why current methods fail to handle large-scale graphs. [Explanation here.](https://openreview.net/forum?id=s6JD_xBS31&amp;noteId=27jNzDx695)
> >
> > In the meantime, we promise to add improvements and analysis of large-scale graphs in the subsequent version.
> >
> > **Thanks for your careful review!**

---

### Author Response · Authors · 2021-08-10
**General Response**

We thank the reviewers for their insightful and constructive feedback. We first answer one issue of concern to many reviewers: **novelty of our model**.

### Contributions and Novelty

Reviewer ZF6P raises an issue of the limited novelty of our model. Here we summarize our contribution as follows.

1. We propose **Laplacian perturbation**, a data augmentation method specifically designed for digraphs. It can generate contrastive views without changing the structure of the digraph, preserving the directed structure information, and minimizing the impact on the subsequent GNN-based encoder.
2. We design an acceleration algorithm that can speed up the computation of Laplacian perturbation by about 100 times. Moreover, we **theoretically**  analyze Laplacian perturbation and give upper and lower bounds on its impact on the digraph.
3. We design **multi-view digraph contrastive learning framework**, which progressively learns from infinite easy-to-hard contrastive views generated by Laplacian perturbation.  It is the **first** framework to leverage multi-task curriculum learning to address the problem of multi-view digraph contrastive learning.

Our model can be divided into two parts, the data augmentation method: **Laplacian perturbation**, and the learning framework: **DiGCL**. Each section has significant improvements on existing methods. Combining the two parts together can produce better results.

We believe these thoughtfully designed methods and in-depth analyses are important contributions to the community of researchers and practitioners working with digraph neural networks. We hope reviewers will support this view.

---

### Author Response · Authors · 2021-09-01
**General response after feedback**

We are grateful to the reviewers for their continuous new thoughts and suggestions. This will be a great encouragement and boost to our subsequent work.

After receiving feedback, we will summarize the reviewers' concerns here and give our views in order to facilitate AC and reviewers to reach a conclusion.


## Experiments on large-scale graph

>First, the first issue raised by reviewer *ZF6P*, *fews*, and *WA6U* is that we need to supplement the experimental results in the case of large-scale datasets.

We would like to clarify one point. **Our contribution is to propose the first contrastive learning method applicable to directed graphs; developing methods applicable to large-scale graphs is not the focus of this paper.**

However, to dispel the reviewers' concerns about the performance of our model with larger-scale data, we added the following experiments as suggested by reviewer ZF6P.

1. **Due to hardware limitations**, we add the performance of our model with other graph contrastive learning methods under a medium-sized dataset (**Amazon-Computer with 287,209 edges**). [Experimental results here.](https://openreview.net/forum?id=s6JD_xBS31&amp;noteId=27jNzDx695)

2. The running time of our **Laplacian perturbation** is compared with other data augmentation schemes in the case of large-scale graph data. Experimental results in Figure 3 (b) of the main text.

3. The running time of our method compared to other **graph contrastive learning frameworks** under different graph scales. [Experimental results here.](https://openreview.net/forum?id=s6JD_xBS31&amp;noteId=27jNzDx695)

4. Analysis of the reasons why current methods fail to handle large-scale graphs. [Explanation here.](https://openreview.net/forum?id=s6JD_xBS31&amp;noteId=27jNzDx695)

## Clarification on the number of views

> Another concern risen by reviewer *WA6U* and  *fews* is the definition of the number of views in the model and point out that using **infinite number of views** can cause conflicts with the definitions of CV, NLP.

We apologize that this description can lead to misunderstandings, but a point of clarification is that **generating continuous (infinite) contrastive views is *not* a contribution of our Laplacian perturbation.** The contributions of our method are as follows.

1. It can generate contrastive views without changing the structure of the digraph, preserving the directed structure information, and minimizing the impact on the subsequent GNN-based encoder.
2. We design an acceleration algorithm that can speed up the computation of Laplacian perturbation by about 100 times. Moreover, we **theoretically** analyze Laplacian perturbation and give upper and lower bounds on its impact on the digraph.

Beyond that, we believe we can take advantage of the larger number of comparison views for the following reasons.

1. The Laplacian perturbation is able to provide as many comparison views as required, and the multi-view contrastive learning framework is also able to learn graph structure information from all generated views.
2. For multi-task curriculum learning, two views will be trained in one subtask, which is consistent with the reviewer's idea, but for the main task, the number of views it can handle is the sum of all subtasks.

### Summary

We will make sure that rebuttal content is included in the updated version based on your recommendations, including improvements and analysis of the large-scale graphs, disambiguation, and more downstream tasks.

In the meantime, we hope that reviewers could turn attention to the **main contribution of our work**, i.e., how to design a contrastive learning framework applicable to digraphs. We believe that introducing unsupervised learning into the more generalized field of digraphs is something that can contribute to the graph machine learning community.

**Thank you!**

---

### Decision · Program_Chairs · 2021-09-28

**Decision:**

Accept (Poster)

**Comment:**

The paper proposes a novel idea of self-supervised learning with digraphs. The scores of the paper are somewhat borderline and the reviewers have had adequate reviews and discussions with the authors. The pros and cons of the paper are well discussed by the reviewers in the reviews. The AC finds the concerns on the novelty and significance of the idea slightly outweigh the pros. The reviewers also pointed out the lack of large-scale experiments and the authors responded that the paper's main contribution is the novel algorithm proposed but not large-scale experiments. This also seems to raise the bar for the theoretical contribution, which the paper does not seem to meet so far.

**Consistency Experiment:**

NeurIPS has a long history of experimentation. In 2014, NeurIPS ran an experiment in which 10% of submissions were reviewed by two independent committees to quantify the randomness in the review process. This year, we repeated a variant of this experiment to see how the quality of the review process has changed over time.  This paper was part of the experiment and was therefore assigned to two committees (consisting of reviewers, an Area Chair, and a Senior Area Chair) that reached independent decisions.  If both committees made the same recommendation, this recommendation was followed. If a single committee recommended acceptance, the paper was accepted (with the exception of a few cases in which the other committee identified what we considered a fatal flaw, e.g., an error in a key result).

This copy’s committee reached the following decision: **Reject**

The other committee assigned to the paper recommended **Accept (Poster)**.  You can find the other set of reviews, along with any follow up discussion with the authors here:
https://openreview.net/forum?id=yLEcG62ANX